# CLImage: Human-Annotated Datasets for Complementary-Label Learning

**Hsiu-Hsuan Wang**[*]                                                        *b09902033@csie.ntu.edu.tw*
*Department of Computer Science and Information Engineering*
*National Taiwan University*

**Tan-Ha Mai**[*]                                                            *d10922024@csie.ntu.edu.tw*
*Department of Computer Science and Information Engineering*
*National Taiwan University*

**Nai-Xuan Ye**                                                             *b09902008@csie.ntu.edu.tw*
*Department of Computer Science and Information Engineering*
*National Taiwan University*

**Wei-I Lin**                                                               *r10922076@csie.ntu.edu.tw*
*Department of Computer Science and Information Engineering*
*National Taiwan University*

**Hsuan-Tien Lin**[†]                                                        *htlin@csie.ntu.edu.tw*
*Department of Computer Science and Information Engineering*
*National Taiwan University*

**Reviewed on OpenReview:** *https://openreview.net/forum?id=FHkWY4aGsN*

## Abstract

Complementary-label learning (CLL) is a weakly-supervised learning paradigm that aims to train a multi-class classifier using only complementary labels, which indicate classes to which an instance does not belong. Despite numerous algorithmic proposals for CLL, their practical applicability remains unverified for two reasons. Firstly, these algorithms often rely on assumptions about the generation of complementary labels, and it is not clear how far the assumptions are from reality. Secondly, their evaluation has been limited to synthetically labeled datasets. To gain insights into the real-world performance of CLL algorithms, we developed a protocol to collect complementary labels from human annotators. Our efforts resulted in the creation of four datasets: CLCIFAR10, CLCIFAR20, CLMicroImageNet10, and CLMicroImageNet20, derived from well-known classification datasets CIFAR10, CIFAR100, and TinyImageNet200. These datasets represent the very first real-world CLL datasets, namely **CLImage**, which are publicly available at: `https://github.com/ntucllab/CLImage_Dataset`. Through extensive benchmark experiments, we discovered a notable decrease in performance when transitioning from synthetically labeled datasets to real-world datasets. We investigated the key factors contributing to the decrease with a thorough dataset-level ablation study. Our analyses highlight annotation noise as the most influential factor in the real-world datasets. In addition, we discover that the biased-nature of human-annotated complementary labels and the difficulty to validate with only complementary labels are two outstanding barriers to practical CLL. These findings suggest that the community focus more research efforts on developing CLL algorithms and validation schemes that are robust to noisy and biased complementary-label distributions.

---

[*]These authors contributed equally to this work.
[†]Corresponding author.

# 1 Introduction

Ordinary multi-class classification methods rely heavily on high-quality labels to train effective classifiers. However, such labels can be expensive and time-consuming to collect in many real-world applications. To address this challenge, researchers have turned their attention towards weakly-supervised learning, which aims to learn from incomplete, inexact, or inaccurate data sources (Zhou, 2018; Sugiyama et al., 2022). This learning paradigm includes but is not limited to noisy-label learning (Frénay & Verleysen, 2014), partial-label learning (Cour et al., 2011), positive-unlabeled learning (Denis, 1998), and complementary-label learning (Ishida et al., 2017).

In this work, we focus on complementary-label learning (CLL). This learning problem involves training a multi-class classifier using only complementary labels, which indicate the classes that a data instance does not belong to. Although several algorithms have been proposed to learn from complementary labels, they were only benchmarked on synthetically labeled datasets with some idealistic assumptions on complementary-label generation (Ishida et al., 2017; 2019; Chou et al., 2020; Wang et al., 2021; Liu et al., 2023). Thus, it remains unclear how well these algorithms perform in practical scenarios.

In particular, current CLL algorithms heavily rely on the *uniform assumption* for generating complementary labels (Ishida et al., 2017), which specifies that complementary labels are generated by uniformly sampling from the set of all possible complementary labels. To alleviate the restrictiveness of the uniform assumption, (Yu et al., 2018) considered a more general *class-conditional assumption*, where the distribution of the complementary labels only depends on its ordinary labels. These assumptions have been used in many subsequent works to generate the *synthetic complementary datasets* for examining CLL algorithms (Ishida et al., 2019; Chou et al., 2020; Feng et al., 2020; Wang et al., 2021; Wei et al., 2022; Liu et al., 2023). Although these assumptions simplify the design and analysis of CLL algorithms, it remains unknown whether these assumptions hold true in practice and whether violation of these assumptions will significantly affect the performance of CLL algorithms. In addition to the uniform or class-conditional assumptions, most existing studies implicitly assumes that the complementary labels are noise-free. That is, they do not mistakenly represent the ordinary labels. While some studies claim to be more robust to noisy complementary labels (Lin & Lin, 2023), they were only tested on synthetic scenarios. It remains unclear how noisy the real-world datasets are, and how such noise affects the performance of current CLL algorithms.

To understand how much the real-world scenario differs from the assumptions, we collect human-annotated complementary-label datasets and conduct benchmarking experiments. We begin by constructing the CLCIFAR10 and CLCIFAR20 datasets, derived from the widely used CIFAR datasets for multi-class classification (Krizhevsky, 2009). Building upon this foundation, we further extend our collection to include two additional human-annotated datasets, CLMicroImageNet10 and CLMicroImageNet20, derived from TinyImageNet200 (Le & Yang, 2015). For all four datasets, we analyze the collected complementary labels, including their noise rates and non-uniform nature. Then, we perform benchmark experiments with diverse state-of-the-art CLL algorithms and conduct dataset-level ablation study on the assumptions of complementary-label generation using the collected datasets.

Our studies reveal annotation noise as the most influential factor affecting the performance of CLL algorithms in real-world datasets. The claim is evidenced by our ablation study results on both synthetic and real-world complementary datasets. Notably, classification accuracy drops from 64.18% on CIFAR10 to 34.85% on CLCIFAR10, highlighting the detrimental impact of noisy complementary labels. To further investigate the role of label noise in the performance gap across different CLL algorithms, we conducted an ablation study that examined various noise levels. The results consistently reinforce our conclusion that the performance disparity between human-annotated complementary labels and synthetically generated complementary labels is primarily driven by label noise. Moreover, through thorough analysis we confirmed that the non-uniform nature of human-annotated complementary labels makes certain CLL algorithms more susceptible to overfitting, although its impact is less pronounced compared to label noise.

These findings immediately suggest that the community focus more research efforts on developing CLL algorithms that are robust to noisy and non-uniform complementary-label distributions. In addition, we used the collected datasets to demonstrate that existing complementary-label-only validation schemes are not

mature yet, suggesting the community a novel research direction for making CLL practical. Our contributions are summarized as follows:

- We designed a collection protocol of complementary labels (CLs) for images, and verified that the protocol collects reasonable human-annotated CLs across different datasets.

- We collected the set of four real-world CL datasets and plan to release **CLImage**[1] to support the continuous research of the community.

- We analyzed the collected datasets with extensive benchmarking experiments, which provides novel and valuable insights for the community.

## 2 Preliminaries on CLL

### 2.1 Complementary-label learning

In ordinary multi-class classification, a dataset $D = \{(\mathbf{x}_i, y_i)\}_{i=1}^n$ that is *i.i.d.* sampled from an unknown distribution is given to the learning algorithm. For each $i$, $\mathbf{x}_i \in \mathbb{R}^M$ represents the $M$-dimension feature of the $i$-th instance and $y_i \in [K] = \{1, 2, \ldots, K\}$ represents the class $\mathbf{x}_i$ belongs to. The goal of the learning algorithm is to learn a classifier from $D$ that can predict the labels of unseen instances correctly. The classifier is typically parameterized by a scoring function $\mathbf{g} \colon \mathbb{R}^M \to \mathbb{R}^K$, and the prediction is made by $\arg\max_{k \in [K]} \mathbf{g}(\mathbf{x})_k$ given an instance $\mathbf{x}$, where $\mathbf{g}(\mathbf{x})_k$ denotes the $k$-th output of $\mathbf{g}(\mathbf{x})$. In contrast to ordinary multi-class classification, CLL shares the same goal of learning a classifier but trains with different labels. In CLL, the ordinary label $y_i$ is not accessible to the learning algorithm. Instead, a complementary label $\bar{y}_i$ is provided, which is a class that the instance $\mathbf{x}_i$ does *not* belong to. The goal of CLL is to learn a classifier that is able to predict the correct labels of unseen instances from a complementary-label dataset $\bar{D} = \{(\mathbf{x}_i, \bar{y}_i)\}_{i=1}^n$.

### 2.2 Common assumptions on CLL

Researchers have made some additional assumptions on the generation process of complementary labels to facilitate the analysis and design of CLL algorithms. One common assumption is the *class-conditional assumption* (Yu et al., 2018). It assumes that the distribution of a complementary label only depends on its ordinary label and is independent of the underlying example's feature, i.e., $P(\bar{y}_i \mid \mathbf{x}_i, y_i) = P(\bar{y}_i \mid y_i)$ for each $i$. One special case of the class-conditional assumption is the *uniform assumption*, which further specifies that the complementary labels are generated uniformly. That is, $P(\bar{y}_i = k | y_i = j) = \frac{1}{K-1}$ for all $k \in [K] \backslash \{j\}$ (Ishida et al., 2017; 2019; Lin & Lin, 2023).

For convenience, a $K \times K$ matrix $T$, called *transition matrix*, is often used to represent how the complementary labels are generated under the class-conditional assumption. $T_{j,k}$ is defined to be the probability of obtaining a complementary label $k$ if the underlying ordinary label is $j$, i.e., $T_{j,k} = P(\bar{y} = k \mid y = j)$ for each $j, k \in [K]$. The diagonals of $T$ hold the conditional probabilities that a complementary label mistakenly represents the ordinary label. That is, they indicate the noise level of the complementary labels. When $T$ contains all zeros on its diagonals, the CLL scenario is called *noiseless*. For instance, the uniform and noiseless assumption can be represented by $T_{j,j} = 0$ for each $j \in [K]$ and $T_{j,k} = \frac{1}{K-1}$ for each $k \neq j$. Class-conditional CLL scenarios based on any other non-uniform $T$ are often called *biased*.

### 2.3 A brief overview of CLL algorithms

The pioneering work by Ishida et al. (2017) studied how to learn from complementary labels under the *uniform assumption* by converting the risk estimator in ordinary multi-class classification to an unbiased risk estimator (**URE**) in CLL (Ishida et al., 2017). **URE** is then found to be prone to overfitting because of negative empirical risks, and is upgraded with two tricks, non-negative risk estimator (**URE-NN**) and gradient accent (**URE-GA**) (Ishida et al., 2019). The *surrogate complementary loss* (**SCL**) algorithm mitigates the overfitting issue of **URE** by a different loss design that decreases the variance of the empirical

---

[1] https://github.com/ntucllab/CLImage_Dataset

estimation. However, these algorithms either rely on the uniform assumption in design or are only tested on the synthetically labeled datasets that obeys the uniform assumption.

To make CLL one step closer to practice, researchers have explored algorithms to go beyond the uniform (and thus noiseless) assumption. (Yu et al., 2018) utilized the forward-correction loss (**FWD**) to accommodate biased complementary label generation by adapting techniques from noisy label learning (Patrini et al., 2017) to change the loss. Additionally, (Gao & Zhang, 2021) proposed the **L-W** algorithm based on discriminatively modeling the distribution of complementary labels through a weighting function, further improving the performance in bias scenario. Furthermore, (Ishiguro et al., 2022) designed robust loss functions for learning from noisy CLs, including **MAE** and **WMAE**, by applying the gradient ascent technique (Ishida et al., 2019) to handle noisy scenarios.

Besides CLL algorithms, a crucial component for making CLL practical is model validation. In ordinary-label learning, this can be done by naively calculating the classification accuracy on a validation dataset. In CLL, this scheme can be intractable if there are not enough ordinary labels. One generic way of model validation is based on the result of (Ishida et al., 2019) by calculating the unbiased risk estimator of the zero-one loss, i.e.,

$$\hat{R}_{01}(\mathbf{g}) = \frac{1}{N} \sum_{i=1}^{N} e_{\overline{y}_i}^{\top} (T^{-1}) \ell_{01}(\mathbf{g}(x_i)) \tag{1}$$

where $e_{\overline{y}_i}$ denotes the one-hot vector of $\overline{y}_i$, $\ell_{01}(\mathbf{g}(x_i))$ denotes the $K$-dimensional vector $(\ell_{01}(\mathbf{g}(x_i), 1), \ldots, \ell_{01}(\mathbf{g}(x_i), K))^T$, and $\ell_{01}(\mathbf{g}(x_i), k) = 0$ if $\arg\max_{k \in [K]} \mathbf{g}(x_i) = k$ and 1 otherwise, representing the zero-one loss of $\mathbf{g}(x_i)$ if the ordinary label is $k$. This estimator will be used in the experiments in Section 6. Another validation objective, surrogate complementary esimation loss (SCEL), was proposed by (Lin & Lin, 2023). SCEL measures the log loss of the complementary probability estimates induced by the probability estimates on the ordinary label space. The formula to calculate SCEL is as follows,

$$\hat{R}_{\text{SCEL}}(\mathbf{g}) = \frac{1}{N} \sum_{i=1}^{N} -\log\left(e_{\overline{y}_i}^{\top} T^{\top} \text{softmax}(\mathbf{g}(x_i))\right). \tag{2}$$

## 3 Construction of the CLImage collection

In this section, we introduce the four complementary-labeled datasets that we collected, CLCIFAR10, CLCIFAR20, CLMicroImageNet10 and CLMicroImageNet20. All datasets are labeled by human annotators on Amazon Mechanical Turk (MTurk)[2].

### 3.1 Datasets and goals

The complementary-labeled datasets are derived from ordinary multi-class classification datasets. CIFAR10, CIFAR100 and TinyImageNet200 (Krizhevsky, 2009; Le & Yang, 2015; Russakovsky et al., 2015). This selection is motivated by the real-world noisy label dataset by (Wei et al., 2022). Building upon the CIFAR and TinyImageNet200 datasets allow us to estimate the noise rate and the empirical transition matrix easily, as they already contain nearly noise-free ordinary labels. In addition, many of the state-of-the-art CLL algorithms have been benchmarked on synthetic complementary labels with the CIFAR datasets (Dosovitskiy et al., 2021; Kolesnikov et al., 2020; Oquab et al., 2024). Our CLCIFAR counterparts immediately allow a fair comparison to those results with the same network architecture.

In addition to our CLCIFAR extensions, we are the first to introduce (Tiny)ImageNet-derived datasets to the CLL literature. Such datasets serve two purposes. First, it allows us to confirm the validity of our collection protocol and findings beyond CIFAR-derived datasets. Second, ImageNet knowingly contains images of higher complexity than CIFAR and can thus be used to challenge the ability of existing CLL algorithms more realistically.

There is a historical note that is worth sharing with the community: We initially attempted to collect complementary labels based on the 100 classes in CIFAR100. But some preliminary testing soon revealed

---

[2]https://www.mturk.com/

that state-of-the-art CLL algorithms cannot produce meaningful classifiers for 100 classes even on synthetic complementary labels that are uniformly and noiselessly generated. We thus set our collection goals to be 10-class classification, which is the focus of most current CLL studies, and 20-class classification, which extends the horizon of CLL and matches the 20 super-class structure in CIFAR.

### 3.2 Complementary label collection protocol

To collect only complementary labels from the CIFAR, TinyImageNet datasets, for each image in the training split, we first randomly sample four distinct labels and ask the human annotators to select any of the *incorrect* one from them. To leave room for analyzing the annotators' behavior, each image is labeled by three different annotators. The four labels are re-sampled for each annotator on each image. That is, each annotator possibly receives a different set of four labels to choose from. An algorithmic description of the protocol is as follows. For each image $\mathbf{x}$,

1. Uniformly sample four labels without replacement from the label set $[K]$.

2. Ask the annotator to select any one of the complementary label $\bar{y}$ from the four sampled labels.

3. Add the pair $(\mathbf{x}, \bar{y})$ to the complementary dataset.

Note that if the annotators always select one of the correct complementary labels uniformly, the empirical transition matrix will also be uniform in expectation. We will inspect the empirical transition matrix in Section 4. The labeling tasks are deployed on MTurk by dividing them into smaller we first divide the total images into smaller human intelligence tasks (HITs). For instance, for constructing the CLCIFAR datasets, we first divide the 50,000 images into five batches of 10,000 images. Then, each batch is further divided into 1,000 HITs with each HIT containing 10 images. Each HIT is deployed to three annotators, who receive 0.03 dollar as the reward by annotating 10 images. To make the labeling task easier and increase clarity, the size of the images are enlarged to $200 \times 200$ pixels.

## 4 Dataset Characteristic

Next, we closely examine the collected complementary labels. We first analyze the error rates of the collected labels, and then verify whether the transition matrix is uniform or not. Finally, we end with an analysis on the behavior of the human annotators observed in the label collection protocol.

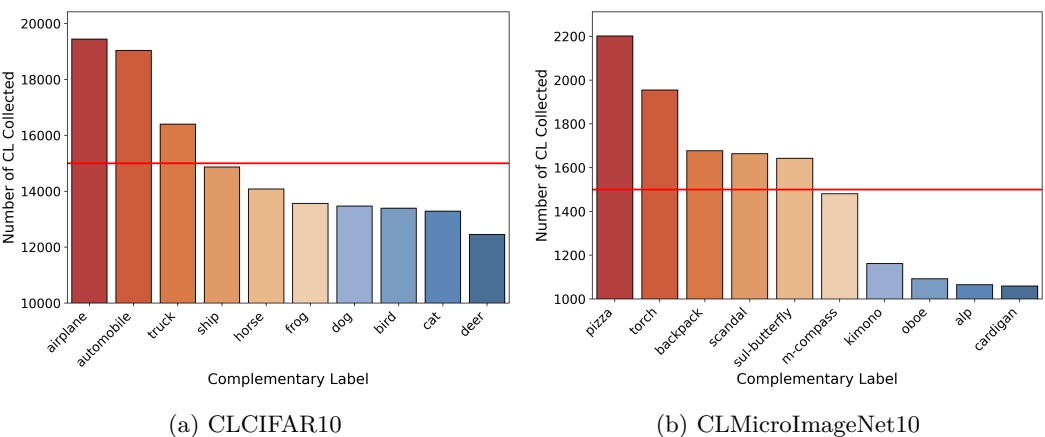

(a) CLCIFAR10        (b) CLMicroImageNet10

Figure 1: The label distribution of CLCIFAR10 and CLMicroImageNet10 datasets.

**Observation 1: noise rate compared to ordinary label collection** We first look at the noise rate of the collected complementary labels. A complementary label is considered to be incorrect if it is actually the ordinary label. The mean error rate made by the human annotators is 3.93% for CLCIFAR10, 2.80% for

CLCIFAR20, 5.19% for CLMicroImageNet10 and 3.21% for CLMicroImageNet20. In theory, we can estimate a random annotator achieves a noise rate of $\frac{1}{K}$ for complementary label annotation and a noise rate of $\frac{K-1}{K}$ for ordinary label annotation. If we compare the human annotators to a random annotator, then for CLCIFAR10, human annotators have 60.7% less noisy labels than the random annotator whereas for CIFAR10-N, human anotators have 78.17% less noisy labels. This demonstrates that human annotators are more competent compared to a random annotator in the ordinary-label annotation. Similarly, human annotators have 44% less noise than a random annotator for CLCIFAR20 and 73.05% less noise for CIFAR100N-coarse (Wei et al., 2022). This observation reveals that while the absolute noise rate is lower in annotating complementary labels, it may be more difficult to be competent against random labels than the ordinary label annotation.

**Observation 2: imbalanced complementary label annotation** Next, we analyze the distribution of the collected complementary labels. The frequency of the complementary labels for the CLCIFAR10 and CLMicroImageNet10 (CLMIN10) datasets are reported in Figure 1. As we can see in the figure, the

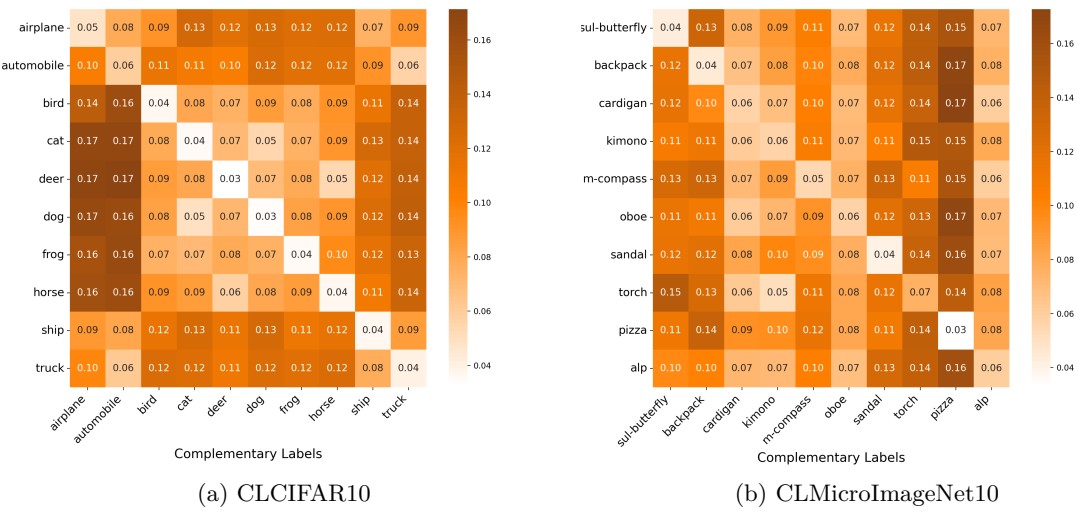

(a) CLCIFAR10         (b) CLMicroImageNet10

Figure 2: The empirical transition matrices of CLCIFAR10 and CLMicroImageNet10.

annotators exhibit specific biases towards certain labels. For instance, in CLCIFAR10, annotators prefer "airplane" and "automobile", while in CLMIN10, they prefer "pizza" and "torch". In CLCIFAR10, the bias is towards labels in different categories, as vehicles ("airplane", "automobile") versus animals ("cat", "bird"). In contrast, in CLMIN10, the bias is towards items that are easily recognizable ("pizza" and "torch") and against those that are less familiar ("cardigan" or "alp").

**Observation 3: biased transition matrix** Finally, we visualize the empirical transition matrix using the collected CLs in Figure 2. Based on the first two observations, we could imagine that the transition matrix is biased. By inspecting Figure 2, we further discover that the bias in the complementary labels are dependent on the true labels. For instance, in CLCIFAR10, despite we see more annotations on airplane and automobile in aggregate, conditioning on the transportation-related labels ("airplane", "automobile", etc), the distribution of the complementary labels becomes more biased towards other animal-related labels ("bird", "cat", etc.) Furthermore, this observation holds true on CLMIN10 as well. Next, we study the impact of the bias and noise on existing CLL algorithms.

We discovered similar patterns in all four human-annotated datasets, validating that our design methodology is practical for collecting real-world CLL image datasets. Due to space limitations, we have included the detailed analysis of CLCIFAR20 and CLMicroImageNet20 in Appendix A.4.

# 5 Experiments

In this section, we benchmarked several state-of-the-art CLL algorithms on CLImage. A significant performance gap between the models trained on the humanly annotated CLCIFAR, CLMicroImageNet dataset and those trained on the synthetically generated complementary labels (CL) was observed in Section 5.1, which motivates us to analyze the possible reasons for the gap with the following experiments. To do so, we discuss the effect of three factors in the label generating process, feature dependency, noise, and biasedness, in Section 5.2, Section 5.3, and Section 5.4, respectively. From our experiment results, we conclude that noise is the dominant factor affecting the performance of the CLL algorithms on CLCIFAR[3].

## 5.1 Standard benchmark on CLImage

**Baseline methods** Several state-of-the-art CLL algorithms were selected for this benchmark. Some of them take the transition matrix $T$ as inputs, which we call $T$**-informed** methods, including two version of forward correction (Yu et al., 2018): **FWD-U** and **FWD-R**, two version of unbiased risk estimator with gradient ascent (Ishida et al., 2019): **URE-GA-U** and **URE-GA-R**, and robust loss (Ishiguro et al., 2022) for learning from noisy CL: **CCE**, **MAE**, **WMAE**, **GCE**, and **SL**[4]. We also included some algorithms that assume the transition matrix $T$ to be uniform, called $T$**-agnostic** methods, including surrogate complementary loss **SCL-NL** and **SCL-EXP** (Chou et al., 2020), discriminative modeling **L-W** and its weighted variant (**L-UW**) (Gao & Zhang, 2021), and pairwise-comparison (**PC**) with the sigmoid loss (Ishida et al., 2017). The details of the algorithms mentioned above are discussed in Appendix B.

**Implementation details** We collected and released three CLs per image to prepare for future studies. However, for this standard benchmark, we chose the first CL from the collected labels for each data instances to form a single CLL dataset, ensuring reproducibility. Then, we trained a ResNet18 (He et al., 2016) model using the baseline methods mentioned above on the single CLL dataset using the Adam optimizer for 300 epochs without learning rate scheduling. Detailed results from the ablation study on various neural network architectures, which further justify our choice of ResNet18 as the backbone, are available in Appendix A.6. The training settings included a fixed weight decay of $10^{-4}$ and a batch size of 512. The experiments were run with Tesla V100-SXM2. For better generalization, we applied standard data augmentation technique, `RandomHorizontalFlip`, `RandomCrop`, and normalization to each image. The learning rate was selected from $\{10^{-3}, 5 \times 10^{-4}, 10^{-4}, 5 \times 10^{-5}, 10^{-5}\}$ using a 10% hold-out validation set. We selected the learning rate with the best classification accuracy on the validation dataset. Note that here we assumed the ordinary labels in the validation dataset are known. We will discuss other validation objectives that rely only on complementary labels in Section 6. As CLL algorithms are prone to overfitting (Ishida et al., 2019; Chou et al., 2020), some previous works did not use the model after training for evaluation. Instead, previous works were performed by evaluating the model on the validation dataset and selecting the epoch with the highest validation accuracy. In this work, we also follow the same aforementioned technique to validate testing set. For reference, we also performed the experiments on synthetically-generated CLL dataset, where the CLs were generated uniformly and noiselessly, denoted uniform-CIFAR.

**Results and discussion** As we can observe in Table 1, there is a significant performance gap between the humanly annotated dataset, CLCIFAR, and the synthetically generated dataset, uniform-CIFAR. The difference between the two datasets can be divided into three parts: (a) whether the generation of complementary labels depends on the feature, (b) whether there is noise, and (c) whether the complementary labels are generated with bias. A negative answer to those questions simplify the problem of CLL. We can gradually simplify CLCIFAR to uniform-CIFAR by chaining those assumptions as follows [5]:

---

[3]Due to space and time constraints, we only provide the results and discussion on the CLCIFAR datasets.

[4]Due to space limitations, we only provided the results of MAE. The remaining results and discussions related to the robust loss methods can be found in Appendix A.3

[4]Note that FWD-R and URE-GA-R assume the empirical transition matrix $T_e$ to be provided. The empirical transition matrix is computed from the labels in the training set, so it is slightly different from a uniform transition matrix $T_u$ in the uniform-CIFAR datasets. As a result, the performances of FWD-R and URE-GA-R do not exactly match those of FWD-U and URE-GA-U, respectively, in the uniform-CIFAR datasets.

[5]The "interpolation" between CLCIFAR and uniform-CIFAR does not necessarily have to be this way. For instance, one can remove the biasedness before removing the noise. We chose this order to reflect the advance of CLL algorithms. First,

Table 1: Standard benchmark results on CLCIFAR/ CLMicroImageNet(CLMIN) and uniform-CIFAR/ MicroImageNet(MIN) datasets. Mean accuracy (± standard deviation) on the testing dataset from four trials with different random seeds. Highest accuracy in each column is highlighted in bold.

| | uniform-CIFAR10 | CLCIFAR10 | uniform-CIFAR20 | CLCIFAR20 | uniform-MIN10 | CLMIN10 | uniform-MIN20 | CLMIN20 |
|---|---|---|---|---|---|---|---|---|
| FWD-U | **64.19**±0.57 | 34.83±0.50 | 21.54±0.37 | 8.03±0.74 | 36.30±1.12 | 23.85±2.76 | 12.57±2.94 | 6.33±1.04 |
| FWD-R | 61.32±0.90 | **38.13**±**0.88** | 21.50±0.38 | **20.27**±**0.53** | 35.70±1.19 | **30.15**±**1.83** | **14.85**±**1.75** | **10.60**±**0.82** |
| URE-GA-U | 50.24±1.11 | 34.72±0.40 | 16.67±1.35 | 10.49±0.52 | 35.70±1.97 | 22.90±2.97 | 11.65±1.90 | 5.75±0.43 |
| URE-GA-R | 50.73±1.83 | 30.23±0.70 | 17.57±0.61 | 6.17±0.82 | 33.65±1.40 | 13.25±5.11 | 9.78±3.88 | 6.50±0.35 |
| SCL-NL | 63.76±0.09 | 34.77±0.60 | 21.37±1.18 | 8.02±0.36 | **37.05**±**1.40** | 21.80±1.85 | 13.00±2.80 | 6.17±0.49 |
| SCL-EXP | 63.29±1.02 | 35.18±0.67 | **21.57**±**1.13** | 7.70±0.41 | 36.55±1.28 | 24.80±1.14 | 12.95±3.38 | 5.58±0.13 |
| L-W | 54.32±0.41 | 32.99±1.01 | 19.59±0.99 | 7.71±0.35 | 33.80±2.66 | 23.80±2.64 | 12.70±2.35 | 6.40±0.29 |
| L-UW | 57.52±0.59 | 34.69±0.32 | 20.71±0.92 | 8.15±0.30 | 35.10±2.74 | 22.40±1.67 | 12.12±3.13 | 6.35±0.86 |
| PC-sigmoid | 37.78±0.80 | 32.15±0.80 | 14.48±0.47 | 12.11±0.46 | 29.10±0.98 | 23.15±0.46 | 10.72±1.38 | 6.90±1.04 |
| ROB-MAE | 59.38±0.63 | 20.23±1.02 | 18.17±1.31 | 5.40±0.59 | 31.50±1.81 | 14.15±0.68 | 6.35±0.86 | 5.38±0.33 |

| | CIFAR10 | CIFAR20 | MIN10 | MIN20 |
|---|---|---|---|---|
| standard supervision | 82.80±0.28 | 63.80±0.49 | 68.70±1.53 | 63.90±1.00 |

$$\boxed{\text{CLCIFAR}} \xRightarrow[\text{Remove feature dependency}]{\text{Section 5.2}} \xRightarrow[\text{Remove noise}]{\text{Section 5.3}} \xRightarrow[\text{Remove biasedness}]{\text{Section 5.4}} \boxed{\text{uniform-CIFAR}}$$

In the following subsections, we will analyze how these three factors affect the performance of the CLL algorithms.

## 5.2 Feature dependency

In this experiment, we verified whether the performance gap resulted from the feature-dependent generation of practical CLs. Conceivably, even if two images belong to the same class, the distribution on the complementary labels could be different. On the other hand, the distributional difference could also be too small to affect model performance, e.g., if $P(\bar{y} \mid y, \mathbf{x}) \approx P(\bar{y} \mid y)$ for most $\mathbf{x}$. Consequently, we decided to further look into whether this assumption can explain the performance gap. To observe the effects of approximating $P(\bar{y} \mid y, \mathbf{x})$ with $P(\bar{y} \mid y)$, we generated two synthetic complementary datasets, CLCIFAR10-*iid* and CLCIFAR20-*iid* by i.i.d. sampling CLs from the empirical transition matrix in CLCIFAR10 and CLCIFAR20, respectively. We proceeded to benchmark the CLL algorithms on CLCIFAR-*iid* and presented the accuracy difference compared to CLCIFAR in Table 2.

**Results and discussion** From Table 2, we observed that the accuracy barely changes on the resampled CLCIFAR-*iid*, suggesting that even if the complementary labels in CLCIFAR could be feature-dependent, this dependency does not affect the model performance significantly. Hence, there might be other factors contributing to the performance gap.

Table 2: Mean accuracy difference (± standard deviation) of different CLL algorithms. A plus indicates the performance on is calculated as CLCIFAR-*i.i.d.* accuracy minus CLCIFAR accuracy.

| | FWD-U | FWD-R | URE-GA-U | URE-GA-R | SCL-NL | SCL-EXP | L-W | L-UW | PC-sigmoid |
|---|---|---|---|---|---|---|---|---|---|
| *CLCIFAR10-iid* | -1.1±2.17 | -0.36±1.15 | -3.03±1.25 | 0.74±0.35 | -0.67±1.81 | -1.97±1.16 | -2.5±0.56 | -3.53±1.36 | -2.03±2.05 |
| *CLCIFAR20-iid* | -0.64±0.39 | -3.53±1.13 | -0.37±0.51 | 1.79±2.34 | -0.28±0.61 | -0.39±0.69 | -0.5±1.37 | -0.82±0.04 | -2.24±0.52 |

## 5.3 Labeling noise

In this experiment, we further investigated the impact of the label noise on the performance gap. Specifically, we measured the accuracy on the noise-removed versions of CLCIFAR datasets, where varying percentages (0%, 25%, 50%, 75%, or 100%) of noisy labels are eliminated.

**Results and discussion** We present the performance of FWD trained on the noise-removed CLCIFAR10 dataset in the left figure in Figure 3. From the figure, we observe a strong positive correlation between

---

researchers address the uniform case (Ishida et al., 2017), then generalize to the biased case (Yu et al., 2018), then consider noisy labels (Ishiguro et al., 2022). There is no work considering feature-dependent complementary labels yet.

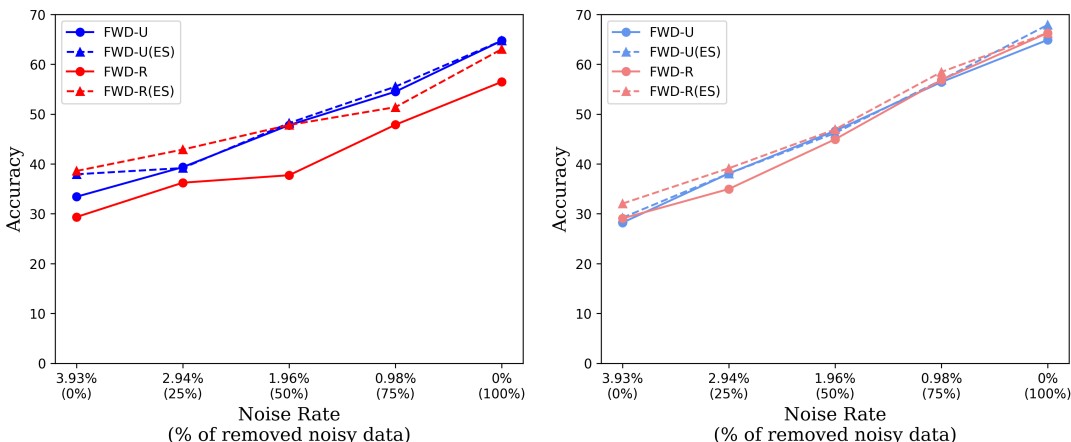

Figure 3: Accuracy of FWD-U and FWD-R on the noise-removed CLCIFAR10 dataset (**Left**) and the uniform-CIFAR10 dataset with uniform noise (**Right**) at varying noise rates.

the performance and the proportion of removed noisy labels. When more noisy labels are removed, the performance gap diminishes and the accuracy approaches that of the ideal uniform-CIFAR dataset. Therefore, we conclude that the performance gap between the humanly annotated CLs and the synthetically generated CLs are primarily attributed to the label noise. The results for FWD-(U/R) and SCL-(NL/EXP) of the noise-removed CLCIFAR10 and CLCIFAR20 datasets are presented in the Figure 4. The other results for other algorithms can be found in Appendix C.

## 5.4 Biasedness of complementary labels

To further study the biasedness of CL as a potential factor contributing to the performance gap, we removed the biasedness from the noise-removed CLCIFAR dataset and examined the resulting accuracy. Specifically, we introduced the same level of uniform noise in uniform-CIFAR dataset and reevaluated the performance of FWD algorithms.

**Results and discussion** The striking similarity between the two curves in the right figure in Figure 3 shows that the accuracy is significantly influenced by label noise, while the biasedness of CL has a negligible impact on the results. Furthermore, we observe that the accuracy difference between the results of the last epoch and the best accuracy of validation set (or early-stopping: **ES**) results becomes smaller when the model is trained on the uniformly generated CLs. That is, the $T$-informed methods are more prone to overfitting when there is a bias in the CL generation.

With the experiment results in Section 5.2, 5.3, and 5.4, we can conclude that the performance gap between humanly annotated CL and synthetically generated CL is primarily attributed to label noise. Additionally, the biasedness of CLs may potentially contribute to overfitting, while the feature-dependent CLs do not detrimentally affect performance empirically. It is worth noting that in the last row of Table 1, the MAE methods that can learn from noisy CL fails to generalize well in the practical dataset. These results suggest that more research on learning with noisy complementary labels can potentially make CLL more realistic.

Following above conclusion, the label noise and biasedness of CL emerge as the two primary factors contributing to overfitting. To gain a better understanding, we conducted deeper investigation into this phenomenon. We demonstrated the necessity of employing data augmentation techniques to prevent overfitting and attempted to address the issue of overfitting by employing an interpolated transition matrix for regularization.

**Ablation on data augmentation** To further investigate the significance of data augmentation, we conducted identical experiments without employing data augmentation during the training phase. As we can observe in the training curves in Figure 5, data augmentation could improve the testing accuracy of all the algorithms we considered.

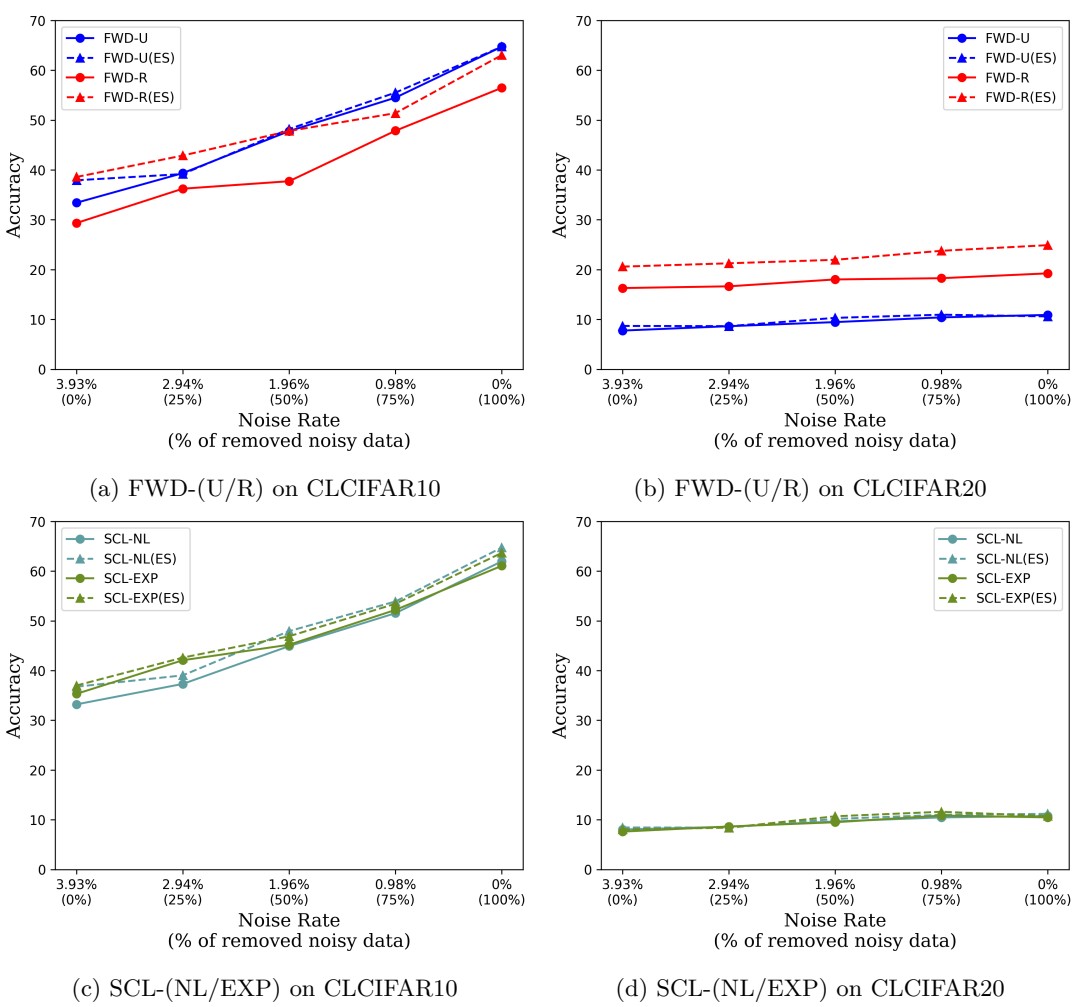

(a) FWD-(U/R) on CLCIFAR10

(b) FWD-(U/R) on CLCIFAR20

(c) SCL-(NL/EXP) on CLCIFAR10

(d) SCL-(NL/EXP) on CLCIFAR20

Figure 4: Accuracy of FWD-(U/R) and SCL-(NL/EXP) on the noise-removed CLCIFAR10 dataset (**Left**) and the CLCIFAR20 dataset with uniform noise (**Right**) at varying noise rates.

Table 3: The overfitting results when there is no data augmentation.

| methods | uniform-CIFAR10 | | CLCIFAR10 | | uniform-CIFAR20 | | CLCIFAR20 | |
|---|---|---|---|---|---|---|---|---|
| | valid_acc | valid_acc (ES) | valid_acc | valid_acc (ES) | valid_acc | valid_acc (ES) | valid_acc | valid_acc (ES) |
| FWD-U | **48.44** | **49.33** | 21.29 | 25.59 | **17.4** | **17.97** | 6.91 | 7.32 |
| FWD-R | - | - | 14.97 | **28.3** | - | - | 6.82 | **14.67** |
| URE-GA-U | 39.55 | 39.67 | 21.0 | 23.53 | 13.52 | 14.08 | 5.55 | 8.38 |
| URE-GA-R | - | - | 19.81 | 20.8 | - | - | 5.0 | 6.43 |
| SCL-NL | 48.2 | 48.27 | **21.96** | 26.51 | 16.55 | 17.54 | **7.1** | 7.92 |
| SCL-EXP | 46.79 | 47.52 | 21.89 | 27.66 | 16.18 | 17.89 | 6.9 | 7.3 |
| L-W | 27.02 | 44.78 | 20.06 | 27.6 | 10.39 | 16.3 | 5.64 | 8.02 |
| L-UW | 31.3 | 46.38 | 20.28 | 26.26 | 12.33 | 16.32 | 6.03 | 8.14 |
| PC-sigmoid | 18.97 | 33.26 | - | - | 7.67 | 10.41 | - | - |

**Ablation on interpolation between $T_u$ and $T_e$** In Table 1, we discovered that the $T$-informed methods did not always deliver better testing accuracy when $T_e$ is given. Looking at the difference between the accuracy of using early-stopping and not using early-stopping, we observe that when the $T_u$ is given to the $T$-informed methods, the difference becomes smaller. This suggests that $T$-informed methods using the empirical transition matrix has greater tendency to overfitting. On the other hand, $T$-informed methods using the uniform transition matrix could be a more robust choice.

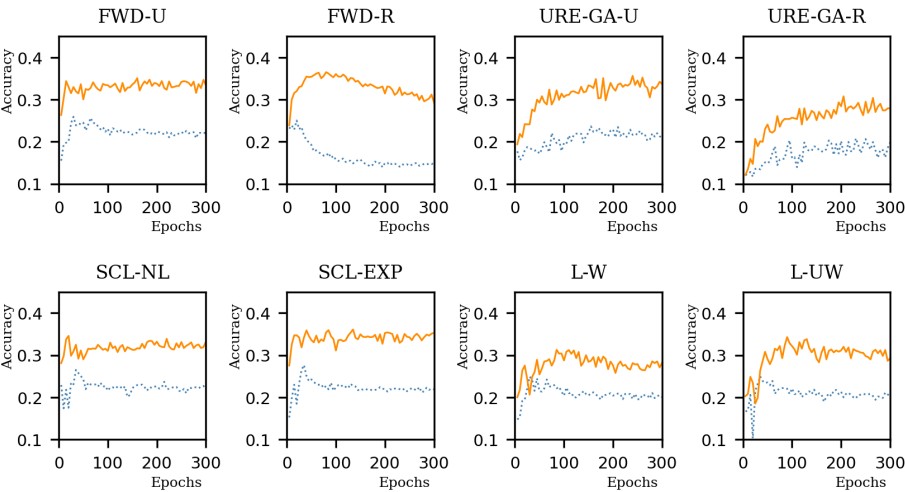

Figure 5: The Overfitting accuracy curve of FWD, URE, SCL-NL, L-W. The dotted line represents the accuracy obtained without data augmentation, while the solid line represents the accuracy with data augmentation included for reference. The accuracy of FWD, SCL-NL, SCL-EXP, L-W, L-UW methods reaches its highest at approximately the 50 epoches and converges to some lower point. The detail numbers are in Table 3

We observe that the uniform transition matrix $T_u$ acts like a regularization choice when the algorithms overfit on CLCIFAR. This results motivate us to study whether we can interpolate between $T_u$ and $T_e$ to let the algorithms utilize the information of transition matrix while preventing overfitting. To do so, we provide an interpolated transition matrix $T_{int} = \alpha T_u + (1 - \alpha)T_e$ to the algorithm, where $\alpha$ controls the scale of the interpolation. As FWD is the $T$-informed method with the most sever overfitting when using $T_u$, we performed this experiment using FWD adn reported the results in Figure 6. As shown in Figure 6, FWD can learn better from an interpolated $T_{int}$, confirming the conjecture that $T_u$ can serve as a regularization role.

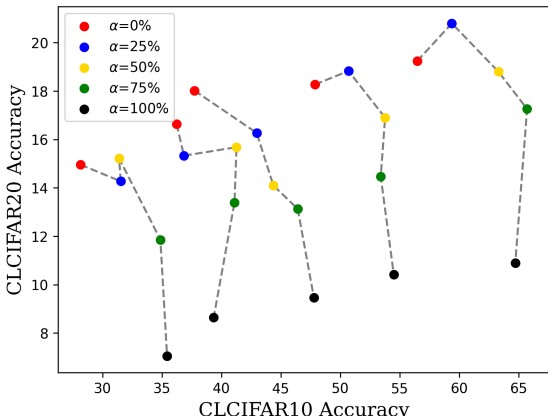

Figure 6: The last epoch accuracy of CLCIFAR10 and CLCIFAR20 for FWD algorithm with an $\alpha$-interpolated transition matrix $T_{int}$. The five solid points on each cruve represent different noise cleaning rate: 0%, 25%, 50%, 75%, 100% from left to right.

## 6 Validation Objectives

Validation is a crucial component in applying CLL algorithms in practice. With the collection of the real-world datasets, we are now able to estimate the difference between using ordinary labels for validation (the common practice in existing CLL studies, as what we do in Section 5) and using complementary labels for validation.

**Validation objectives** As discussed in Section 2, validating the model performance solely with complementary labels poses a non-trivial challenge. To the best of our knowledge, only two existing CLL studies offer some possibility to evaluate a classifier *with only complementary labels*. They are URE (Ishida et al., 2019) and SCEL (Lin & Lin, 2023). We take these two validation objectives to select the optimal learning rate from $\{10^{-3},\ 5\times 10^{-4},\ 10^{-4},\ 5\times 10^{-5},\ 10^{-5}\}$ and provides the accuracy on testing set in Table 5. We compare the result to another validation objective that computes the accuracy on an equal number of *ordinary labels*. Our goal was to determine the gap between using complementary labels and ordinary labels for validation. We selected the best learning rate based on the validation objectives for URE, SCEL, and ordinary-label accuracy, and then report the test performance, as shown in Table 4 for synthetically labeled datasets and Table 5 for real-world datasets.

Table 4: The testing accuracy of models evaluated with URE and SCEL.

| | uniform-CIFAR10 | | | | uniform-CIFAR20 | | | | uniform-MIN10 | | | | uniform-MIN20 | | | |
|---|---|---|---|---|---|---|---|---|---|---|---|---|---|---|---|---|
| | URE | SCEL | valid acc | gap (↓) | URE | SCEL | valid acc | gap (↓) | URE | SCEL | valid acc | gap (↓) | URE | SCEL | valid acc | gap (↓) |
| FWD-U | 53.41±5.51 | 50.36±3.25 | 64.19±0.57 | 10.78 | 16.73±2.29 | 16.52±2.61 | 21.54±0.37 | 4.81 | 33.65±2.84 | 33.20±3.16 | 36.30±1.12 | 2.65 | 10.10±2.66 | 9.15±1.68 | 12.57±2.94 | 2.47 |
| FWD-R | 52.55±4.06 | 49.17±3.11 | 61.32±0.90 | 8.77 | 18.29±0.39 | 16.61±2.65 | 21.50±0.38 | 3.21 | 32.15±3.40 | 33.10±2.03 | 35.70±1.19 | 2.60 | 12.72±3.28 | 11.57±2.91 | 14.85±1.75 | 2.12 |
| URE-GA-U | 48.68±1.11 | 49.29±1.67 | 50.24±1.11 | 0.95 | 15.23±2.35 | 16.09±1.23 | 16.67±1.35 | 0.58 | 28.10±5.24 | 34.35±2.39 | 35.70±1.97 | 1.35 | 8.53±1.55 | 8.52±1.38 | 11.65±1.90 | 3.12 |
| URE-GA-R | 50.49±1.21 | 50.25±1.57 | 50.73±1.83 | 0.25 | 15.68±1.35 | 16.12±0.95 | 17.57±0.61 | 1.45 | 29.85±4.73 | 34.10±1.90 | 33.65±1.40 | -0.45 | 7.15±2.13 | 7.12±2.42 | 9.78±3.88 | 2.63 |
| SCL-NL | 54.32±6.71 | 51.03±3.12 | 63.76±0.09 | 9.44 | 15.65±3.06 | 16.32±3.11 | 21.37±1.18 | 5.05 | 32.95±3.13 | 33.20±3.69 | 37.05±1.40 | 3.85 | 11.50±3.76 | 9.28±2.55 | 13.00±2.80 | 1.50 |
| SCL-EXP | 50.98±6.83 | 41.61±3.52 | 63.29±1.02 | 12.30 | 16.71±2.72 | 16.15±2.55 | 21.57±1.13 | 4.86 | 32.95±2.91 | 29.70±2.83 | 36.55±1.28 | 3.60 | 10.53±2.02 | 8.83±3.19 | 12.95±3.38 | 2.43 |
| L-W | 46.88±9.44 | 50.36±0.47 | 54.32±0.41 | 3.95 | 16.26±1.93 | 14.67±1.59 | 19.59±0.99 | 3.33 | 17.70±9.90 | 28.60±5.15 | 33.80±2.66 | 5.20 | 8.58±1.25 | 7.70±0.35 | 12.70±2.35 | 4.12 |
| L-UW | 52.47±3.63 | 51.15±1.61 | 57.52±0.59 | 5.05 | 16.10±1.51 | 15.58±1.97 | 20.71±0.92 | 4.62 | 26.60±7.14 | 25.60±7.14 | 35.10±2.74 | 9.50 | 10.60±2.36 | 8.28±2.02 | 12.12±3.13 | 1.52 |
| PC-sigmoid | 35.29±1.67 | 34.82±1.24 | 37.78±0.80 | 2.49 | 13.41±0.95 | 13.40±0.72 | 14.48±0.47 | 1.07 | 25.55±5.99 | 27.05±5.66 | 29.10±0.98 | 2.05 | 7.75±1.73 | 8.72±0.06 | 10.72±1.38 | 2.00 |
| ROB-MAE | 57.99±1.72 | 57.79±2.03 | 59.38±0.63 | 1.39 | 17.07±2.02 | 15.62±1.79 | 18.17±1.31 | 1.11 | 30.15±4.22 | 29.15±2.90 | 31.50±1.81 | 1.35 | 5.42±0.27 | 5.03±0.54 | 6.35±0.86 | 0.92 |

Table 5: The testing accuracy of models evaluated with URE and SCEL.

| | CLCIFAR10 | | | | CLCIFAR20 | | | | CLMIN10 | | | | CLMIN20 | | | |
|---|---|---|---|---|---|---|---|---|---|---|---|---|---|---|---|---|
| | URE | SCEL | valid acc | gap (↓) | URE | SCEL | valid acc | gap (↓) | URE | SCEL | valid acc | gap (↓) | URE | SCEL | valid acc | gap (↓) |
| FWD-U | 33.13±1.30 | 31.86±1.52 | 34.83±0.50 | 1.70 | 6.70±0.46 | 7.10±0.48 | 8.03±0.74 | 0.93 | 20.75±2.12 | 20.20±0.72 | 23.85±2.76 | 3.10 | 4.97±0.72 | 4.55±0.81 | 6.33±1.04 | 1.35 |
| FWD-R | 33.70±3.38 | 35.64±1.37 | 38.13±0.88 | 2.49 | 17.35±2.32 | 18.40±1.56 | 20.27±0.53 | 1.86 | 22.15±4.15 | 29.15±1.93 | 30.15±1.83 | 1.00 | 8.60±1.32 | 9.90±1.19 | 10.60±0.82 | 0.70 |
| URE-GA-U | 30.45±3.58 | 33.21±1.12 | 34.72±0.40 | 1.51 | 7.03±0.61 | 8.71±0.74 | 10.49±0.52 | 1.79 | 17.05±3.35 | 21.30±3.01 | 22.90±2.97 | 1.60 | 4.27±0.80 | 5.03±0.48 | 5.75±0.43 | 0.72 |
| URE-GA-R | 27.39±1.89 | 28.32±1.38 | 30.23±0.70 | 1.91 | 3.58±0.47 | 5.42±0.96 | 6.17±0.82 | 0.75 | 8.90±1.03 | 10.30±1.53 | 13.25±5.11 | 2.95 | 5.15±0.62 | 5.57±1.54 | 6.50±0.35 | 0.93 |
| SCL-NL | 33.55±0.79 | 33.70±1.33 | 34.77±0.60 | 1.07 | 6.73±0.51 | 7.47±0.56 | 8.02±0.36 | 0.55 | 19.55±1.37 | 22.15±1.76 | 21.80±1.85 | -0.35 | 4.83±1.12 | 5.20±0.51 | 6.17±0.49 | 0.98 |
| SCL-EXP | 31.30±2.62 | 33.47±1.16 | 35.18±0.67 | 1.71 | 6.83±0.23 | 7.03±0.62 | 7.70±0.41 | 0.66 | 18.35±1.60 | 20.65±1.39 | 24.80±1.14 | 4.15 | 5.05±0.56 | 4.45±0.74 | 5.58±0.13 | 0.52 |
| L-W | 27.49±4.30 | 30.32±2.40 | 32.99±1.01 | 2.67 | 5.90±0.29 | 7.18±0.31 | 7.71±0.35 | 0.53 | 19.30±4.66 | 18.95±2.30 | 23.80±2.64 | 4.50 | 5.97±0.33 | 5.55±0.17 | 6.40±0.29 | 0.43 |
| L-UW | 28.90±2.01 | 29.78±2.69 | 34.69±0.32 | 4.91 | 6.40±0.42 | 8.16±0.30 | 8.15±0.30 | -0.01 | 18.25±4.31 | 19.80±1.61 | 22.40±1.67 | 2.60 | 5.82±0.77 | 6.48±1.03 | 6.35±0.86 | -0.13 |
| PC-sigmoid | 24.83±5.94 | 31.48±1.93 | 32.15±0.80 | 0.67 | 7.98±2.47 | 10.59±0.87 | 12.11±0.46 | 1.51 | 12.55±1.31 | 17.85±4.61 | 23.15±0.46 | 5.30 | 6.40±1.19 | 5.33±1.28 | 6.90±1.04 | 0.50 |
| ROB-MAE | 18.80±1.64 | 18.75±0.99 | 20.23±1.02 | 1.43 | 4.70±0.43 | 4.87±0.32 | 5.40±0.59 | 0.53 | 11.80±2.92 | 14.35±1.59 | 14.15±0.68 | -0.20 | 5.08±0.44 | 4.62±0.66 | 5.38±0.33 | 0.30 |

**Results and discussion** Firstly, there appears no clear winner between URE and SCEL, both using only CLs for validation. Validating with the ordinary-label accuracy generally provides stronger performance than URE/SCEL, and the test performance gap between validating with ordinary labels and validating with complementary labels can be as big as nearly 5%. These findings suggest that using purely complementary labels for validation, whether through URE or SCEL, still suffers from a non-negligible performance drop compared to using ordinary validation. That is, the numbers reported in existing studies, which validates with ordinal labels, can be optimistic for practice. Whether this gap can be further reduced remains an open research problem and the community can pay more attention on that to make CLL more practical.

# 7 Alternative to Current Data Collection Protocol

In this work, we used a specific protocol to collect complementary labels, but we acknowledge there are several alternative protocols that could also be considered. Here we highlight possibilities while explaining the rationale behind our current design choices.

The first alternative would be to provide annotators with more than four classes. However, realistically, increasing the number beyond four would significantly increase the effort required from annotators. We chose four because it is a common standard for multiple-choice scenarios and was effectively employed in prior research (Wei et al., 2022) that used human annotations to collect noisy labels in real-world settings. Nevertheless, we admit exploring different numbers of labels beyond four presents an interesting area for future research in complementary dataset.

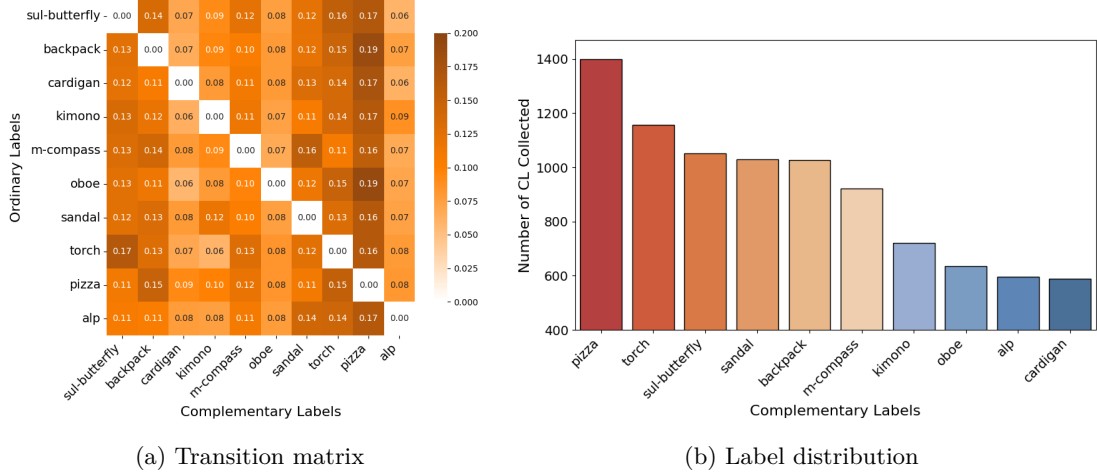

(a) Transition matrix

(b) Label distribution

Figure 7: The transition matrix (**a**) and label distribution (**b**) of CLMicroImageNet10 with the true labels hidden.

A second alternative, we considered was using a simple "yes/no" format. However, this approach posed two main issues. First, it would generate a dataset containing both true and complementary labels, which conflicts with our goal of creating a purely complementary dataset. Second, the resulting dataset would likely contain increased noise, complicating clear analysis and validation of the label transition assumptions.

Another possible protocol involves explicitly hiding the true label during annotation to minimize noise. However, this approach is less realistic since, in real-world scenarios, true labels are typically unknown (otherwise complementary annotation will not be necessary). Note that part of our current dataset, which does not have choices that include the true label, can be taken to analyze the effect of such a protocol. We show in Figure 7) that such a protocol does not lead to much change of the transition matrix (except for the diagonal noise removal) and label distribution, when compared with our original choice of protocol.

We view our work as an initial exploration into collecting complementary labels, and we hope that this work will inspire the future work on other possible collection protocols of complementary label.

## 8 Conclusion

In this paper, we devised a protocol to collect complementary labels from human annotators. Utilizing this protocol, we curated four real-world datasets, CLCIFAR10, CLCIFAR20, CLMicroImageNet10, and CLMicroImageNet20 and made them publicly available to the research community. Through our meticulous analysis of these datasets, we confirmed the presence of noise and bias in the human-annotated complementary labels, challenging some of the underlying assumptions of existing CLL algorithms. Extensive benchmarking experiments revealed that noise is a critical factor that undermines the effectiveness of most existing CLL algorithms. Furthermore, the biased complementary labels can trigger overfitting, even for algorithms explicitly designed to leverage this bias information. In addition, our study on the validation objective for CLL suggests that validating with only complementary labels causes significant performance degrading. These findings emphasize the need for the community to dedicate more effort on those issues. The curated datasets pave the way for the community to create more practical and applicable CLL solutions.

## 9 Limitations

To ensure the compatibility with previous CLL algorithms, our work focuses on image datasets based on CIFAR10/100, and TinyImageNet. It is worth investigating the real-world CLL datasets on larger datasets, such as ImageNet, and other domains. On the other hand, the proposed protocol focuses on collecting real-world complementary labels for analyzing the common assumptions on CLL. That said, it is also crucial to understand efficient ways to collect complementary labels in practice, e.g., by asking annotators binary

questions to collect ordinary and complementary labels simultaneously. We leave these directions as future works and hope that our work can open the way for the community to understand these questions.

**Broader Impact Statement**

It is known that weakly-supervised learning can be used for some privacy-preserving applications. While our CLImage datasets do not fall under such privacy-preserving applications, we suggest that practitioners exercise caution when exploring these use cases.

**Acknowledgement**

We thank the anonymous reviewers and members of CLLab for their constructive feedback. This work is supported by the National Science and Technology Council in Taiwan via NSTC 113-2628-E-002-003, NSTC 113-2634-F-002-008 and NSTC 112-2628-E-002-030. We thank to National Center for High-performance Computing (NCHC) of National Applied Research Laboratories (NARLabs) in Taiwan for providing computational and storage resources.

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

# Appendix

## A More discussion on practical noise and extended ablation study

Our work found out that the labeling noise is the main factor contributing to the performance gap between synthetic CL and practical CL. Hence, we conducted deeper investigation into some directions to handle the practical noise. In Section A.1, we discussed the performance improvement when more human-annotated complementary labels were available. In Section A.2, we designed the synthetic CLCIFAR-N dataset to study the difference between synthetic uniform noise and practical noise. In Section A.3, we provided the benchmark results of all robust loss methods to emphasize the essence of studying a practical complementary label dataset. In Section A.4, we discussed result analysis of CLCIFAR20 and CLMicroImageNet20 datasets and described the process how MicroImageNet10 and MicroImageNet20 datasets were generated in Section A.5. In Section A.6, we provided an ablation study on different network architectures commonly used in complementary label learning.

### A.1 Multiple complementary labels

In this experiment, we studied the case when there were multiple CLs for a data instance. We duplicated the data instance and assigned them with another practical label from the annotators. The results of this experiment were summarized in Table 6.

For CLCIFAR10, we observe that the model achieved better learning performance when trained on data instances with more CLs. However, the issue of overfitting persists even with the increased number of labels. In the case of CLCIFAR20, we found that without employing early stopping techniques, it is challenging to achieve improved results as the number of labels increased. Furthermore, the overfitting problem becomes more pronounced with the increased number of labels. Overall, these findings shed light on the challenges posed by multiple CLs and the persistence of overfitting.

### A.2 Benchmarks with synthetic noise

**Generation process of CLCIFAR-N** Inspired by the conclusions drawn in Section 5.3, we investigated another avenue of research: the generalization capabilities of methods when transitioning from synthetically labeled datasets with uniform noise to practical datasets. To obtain a general synthetic dataset with minimum assumption, we introduced CLCIFAR-N. This synthetic dataset contains unifrom CL and uniform real world noise from CLCIFAR dataset. The complementary labels of CLCIFAR-N are *i.i.d.* sampled from $T_{syn}$, where the diagonal entries are set to be $3.93\%/10$ (for generating CL for CIFAR10) or $2.8\%/20$ (for generating CL for CIFAR20). The non-diagonal entries are uniformly distributed. This construction allows us to generate a synthetic dataset that mimics real-world scenarios more closely with minimum knowledge.

**Benchmark results** We ran the benchmark experiments with the identical settings as in Section 5.1 and present the results in Table 7. The performance difference between sythetic noise and practical noise are illustrated in the *diff* columns. A smaller difference indicates a better generalization capability of the models. Interestingly, the robust loss methods exhibit superiority on the synthetic CLCIFAR10-N dataset but struggle to generalize well on real-world datasets. This finding suggests the existence of fundamental differences between synthetic noise and practical noise. Further investigation into these differences is left as an avenue for future research.

### A.3 Results of the robust loss methods

The original design of the robust loss aims to obtain the optimal risk minimizer even in the presence of corrupted labels. However, their methods do not generalized well on practical datasets. The results are provided in Table 8. In other words, solely considering synthetic noisy CLs does not guarantee performance on real-world datasets. These results once again underscore the importance of the CLCIFAR dataset.

Table 6: Learning with Multiple CL: The figure shows the classification accuracy of each task with early stopping indicated in brackets. The highest accuracy in each column is bolded for ease of comparison.

| | CLCIFAR10 | | | CLCIFAR20 | | |
|---|---|---|---|---|---|---|
| num CL | 1 | 2 | 3 | 1 | 2 | 3 |
| FWD-U | 34.09(36.83) | 41.95(41.53) | 42.88(45.18) | 7.47(8.27) | 8.28(8.78) | 8.15(10.27) |
| FWD-R | 28.88(**38.9**) | 34.33(**47.07**) | 37.84(**49.76**) | **16.14**(**20.31**) | **16.99**(**23.41**) | 15.54(**24.19**) |
| URE-GA-U | **34.59**(36.39) | **45.71**(44.85) | **45.97**(47.97) | 7.59(10.06) | 8.42(11.52) | 8.53(12.75) |
| URE-GA-R | 28.7(30.94) | 42.73(43.34) | 44.73(47.36) | 5.24(5.46) | 6.77(6.92) | 5.0(5.55) |
| SCL-NL | 33.8(37.81) | 40.67(42.58) | 43.39(45.2) | 7.58(8.53) | 6.77(6.92) | 5.0(5.55) |
| SCL-EXP | **34.59**(36.96) | 40.89(42.99) | 44.4(47.9) | 7.55(8.11) | 7.42(8.39) | 8.0(9.31) |
| L-W | 28.04(34.55) | 34.96(41.83) | 39.05(47.46) | 7.08(8.74) | 8.06(8.76) | 8.03(10.18) |
| L-UW | 30.63(35.13) | 38.05(43.32) | 39.49(45.82) | 7.36(8.71) | 7.03(8.55) | 7.86(10.11) |
| PC-sigmoid | 24.38(35.88) | 25.63(39.82) | 33.89(43.75) | 9.27(14.26) | 11.91(16.07) | **17.68**(14.13) |

Table 7: Benchmark results on CLCIFAR-N datasets. The classification accuracy difference is calculated by subtracting the practical CLCIFAR dataset from the performance on the synthetic CLCIFAR-N dataset.

| | CLCIFAR10-N | diff($\downarrow$) | CLCIFAR20-N | diff($\downarrow$) |
|---|---|---|---|---|
| FWD-U | 37.1 | 2.2 | **7.58** | 0.11 |
| FWD-R | - | - | - | - |
| URE-GA-U | 31.29 | -3.3 | 8.1 | 0.5 |
| URE-GA-R | - | - | - | - |
| SCL-NL | 37.79 | 2.06 | 7.75 | 0.16 |
| SCL-EXP | 35.86 | 3.19 | 6.95 | -0.59 |
| L-W | 30.1 | 2.06 | 6.16 | -0.91 |
| L-UW | 32.69 | 2.05 | 6.89 | -0.47 |
| PC-sigmoid | 19.64 | -4.73 | 6.54 | -2.72 |
| CCE | 32.34 | 13.45 | 5.71 | 0.71 |
| MAE | **41.34** | 23.09 | 6.83 | 1.83 |
| WMAE | 37.62 | 22.26 | 6.36 | 1.08 |
| GCE | 35.00 | 18.71 | 6.7 | 1.7 |
| SL | 29.98 | 12.29 | 6.08 | 1.05 |

## A.4 Result analysis of CLCIFAR20 and MicroImageNet20

In this section, we further investigate the complementary labels collected from the CLCIFAR20 and MicroImageNet20 datasets. We followed similar observation and analyzed in the Section 4. Our observation and analysis are described as below:

**Observation 1: noise rate compared to ordinary label collection** We observed that the noise rates for the complementary labels collected from the CLCIFAR20 and MicroImageNet20 datasets are 2.80% and 3.21%, respectively. This finding is consistent with the observations discussed in Section 4. The lower noise rate in the CLCIFAR20 dataset compared to MicroImageNet20 can be attributed to the greater difficulty in labeling the MicroImageNet20 dataset.

**Observation 2: imbalanced complementary label annotation** Next, we analyzed the distribution of the collected complementary labels. The frequencies of these labels for the CLCIFAR20 and CLMicroImageNet20 (CLMIN20) datasets are shown in Figure 8. The figure reveals that annotators exhibit specific biases towards certain labels. For example, in CLCIFAR20, annotators show a preference for labels such as "fish", "flowers", "people", "trees", "food container", and "transportation vehicles". In CLMIN20 [6], they favor "iPod" and "tractor". In CLCIFAR20, the bias tends towards labels with shorter, more concrete, and understandable names. Conversely, in CLMIN20, the preference is for easily recognizable items as "iPod", and "tractor", while less familiar items such as "bannister", "american lobste", "snorkel", and "gazelle" are less favored.

**Observation 3: biased transition matrix** Finally, we visualized the empirical transition matrix using the collected complementary labels, as shown in Figure 9. Our observations indicate that the transition matrix

---

[6]The mapping between class numbers and labels for CLCIFAR10 and CLMIN20 is provided in Appendix E.

Table 8: Standard benchmark results on CLCIFAR and uniform-CIFAR datasets for the robust loss method. Mean accuracy (± standard deviation) on the testing dataset from four trials with different random seeds. Highest accuracy in each column is highlighted in bold.

| | uniform-CIFAR10 | | CLCIFAR10 | | uniform-CIFAR20 | | CLCIFAR20 | |
|---|---|---|---|---|---|---|---|---|
| methods | valid_acc | valid_acc (ES) | valid_acc | valid_acc (ES) | valid_acc | valid_acc (ES) | valid_acc | valid_acc (ES) |
| CCE | 46.57±1.75 | 49.51±0.73 | 16.18±2.97 | 20.18±3.39 | 12.54±0.40 | 14.62±1.29 | 5.07±0.05 | 5.41±0.30 |
| MAE | 57.37±0.48 | 58.50±0.97 | 16.30±2.27 | 19.44±4.41 | 16.72±1.52 | 17.63±1.63 | 5.11±0.11 | 5.87±0.26 |
| WMAE | - | - | 13.01±1.89 | 15.51±0.75 | - | - | 5.31±0.27 | 6.65±0.65 |
| GCE | 58.10±1.54 | 59.44±2.30 | 14.31±1.44 | 18.97±2.16 | 15.86±1.93 | 17.09±1.19 | 5.21±0.29 | 5.76±0.32 |
| SL | 41.13±1.64 | 42.64±0.11 | 16.45±2.80 | 19.28±3.16 | 13.60±0.55 | 15.70±1.23 | 5.44±0.29 | 6.59±0.43 |

is biased. Specifically, we discovered that the bias in the complementary labels is dependent on the true labels, as depicted in Figure 9. In CLCIFAR20, there are more annotations for labels with shorter, more concrete, and understandable names, such as "fish", "flowers", "people", and "transportation vehicles". This results in a distribution that is more biased towards these labels. A similar pattern of bias is observed in CLMIN20, where annotators favored easily recognizable items like "iPod" and "tractor", while less familiar items received fewer annotations.

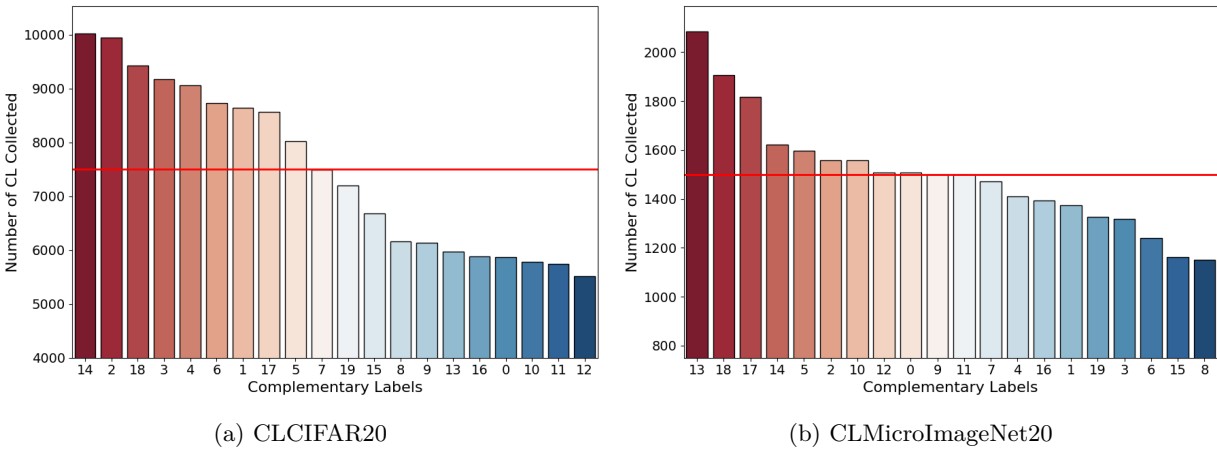

(a) CLCIFAR20 (b) CLMicroImageNet20

Figure 8: The label distribution of CLCIFAR20 and CLMicroImageNet20 datasets. The full label names are provided in Appendix E.

### A.5 MicroImageNet dataset generation

To generate the MicroImageNet10 and MicroImageNet20 datasets, we began by randomly selecting 10 classes from the 200 available in MicroImageNet to create MicroImageNet10. Similarly, we randomly selected 20 classes to form MicroImageNet20. The selected classes are listed in Table 11 of Appendix E. Each class in the TinyImageNet200 dataset contains multiple labels. To ensure reproducibility and facilitate human annotation, we chose the first label to represent the primary label of each class, as detailed in Appendix E. Each class in the MicroImageNet10/20 datasets comprises 500 images for the training set and 50 images for the validation set. To collect complementary labels for the MicroImageNet10/20 datasets, we followed a protocol similar to the one described in Section 3.2.

### A.6 Result analysis of different neural network architectures

We chose to begin benchmarking with simpler models due to the known issue of overfitting in CLL (Chou et al., 2020). Simpler models are typically less prone to overfitting, making them suitable for initial benchmarking. We conducted an ablation study comparing different neural network architectures, including ResNet18, ResNet34, ResNet50, and DenseNet121, to determine the most suitable architecture for our complementary

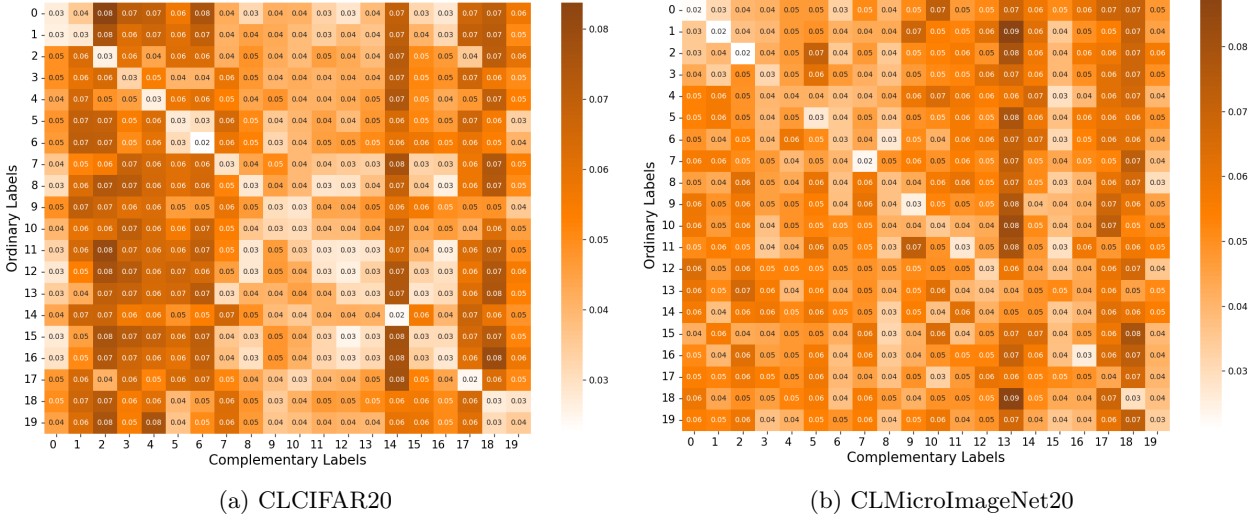

(a) CLCIFAR20          (b) CLMicroImageNet20

Figure 9: The empirical transition matrices of CLCIFAR20 and CLMicroImageNet20. The label names of CLCIFAR20 and CLMicroImageNet20 are abbreviated as indexes to save space. The full label names are provided in Appendix E.

dataset. Results shown in Table 9 indicate that ResNet18 achieved the best performance. Additionally, ResNet18 closely aligns with ResNet architectures commonly used in CLL studies (Ye et al., 2024; Xu et al., 2020). Thus, we selected ResNet18 for benchmarking purposes.

Table 9: Performance of various neural network architectures using SCL-NL and FWD-R algorithms on our real-world complementary datasets. The highest accuracy in each column is highlighted in bold.

| | CLCIFAR10 | | | | CLCIFAR20 | | | | CLMIN10 | | | | CLMIN20 | | | |
|---|---|---|---|---|---|---|---|---|---|---|---|---|---|---|---|---|
| | ResNet18 | ResNet34 | ResNet50 | DenseNet121 | ResNet18 | ResNet34 | ResNet50 | DenseNet121 | ResNet18 | ResNet34 | ResNet50 | DenseNet121 | ResNet18 | ResNet34 | ResNet50 | DenseNet121 |
| **SCL-NL** | **34.77** | 25.13 | 24.55 | 31.42 | **8.02** | 5.62 | 7.45 | 7.98 | **21.80** | 12.60 | 10.40 | 11.40 | **6.17** | 5.00 | 3.60 | 5.10 |
| **FWD-R** | **38.12** | 27.60 | 26.06 | 27.09 | **20.27** | 12.21 | 13.78 | 14.26 | **30.15** | 15.20 | 21.00 | 15.80 | **10.60** | 5.80 | 6.80 | 6.50 |

# B   An overview of the complementary-label learning algorithms

In this section, we review the algorithms benchmarked in Section 5.

## B.1   T-informed CLL algorithms

Some of them take the transition matrix $T$ as inputs, which we call $T$-informed methods, including

- Two versions of forward correction method (Yu et al., 2018): **FWD-U** and **FWD-R**. They utilize a uniform transition matrix $T_u$ and an empirical transition matrix $T_e$ as input, respectively.

- Two versions of unbiased risk estimator with gradient ascent (Ishida et al., 2019): **URE-GA-U** with a uniform transition matrix $T_u$ and **URE-GA-R** with an empirical transition matrix $T_e$.

- Robust loss methods (Ishiguro et al., 2022) for learning from noisy CL, including **CCE**, **MAE**, **WMAE**, **GCE**, and **SL**[7]. We applied the gradient ascent technique (Ishida et al., 2019) as recommended in the original paper.

---

[7]Due to space limitations, we only provided the results of MAE. The remaining results and discussions related to the robust loss methods can be found in Appendix A.3.

In practice, the empirical transition matrix $T_e$ is not accessible to the learning algorithm, but we assume that the correct $T_e$ is given to **FWD-R**, **URE-GA-R** and the robust loss methods for simplicity.

T-informed CLL algorithms are those that has the transition matrix as inputs, includes but not limited to Forward loss correction (FWD) and Unbiased risk estimate (URE). They are expected to utilize the information of the transition matrix to provide better performance when the complementary labels are not generated uniformly. The transition matrix, however, may not be accessible in practice. In this case, a uniform transition matrix $T_u$ is typically provided to the algorithms as a default choice. In the benchmark in Section 5, we considered both scenarios in which the empirical transition matrix $T_e$ or the uniform transition matrix $T_u$ was provided.

**FWD**  Forward loss correction utilizes the information of a transition matrix $T$ in its loss function as in Eq. 3 (Yu et al., 2018). Essentially, this method trains model $f$ by minimizing the following loss function.

$$R(\mathbf{g}) = \frac{1}{N} \sum_{i=1}^{N} \ell(T^\top \text{softmax}(\mathbf{g}(x_i)), \bar{y}_i) \tag{3}$$

where $T$ is the transition matrix provided to the method. We use **FWD-U** and **FWD-R** to indicate the cases that $T$ equals $T_u$ and $T_e$, respectively.

**URE-GA**  (Ishida et al., 2017) proposed an unbiased risk estimator (URE) for learning from complementary label. The loss of the URE is defined as follows,

$$R(\mathbf{g}) = \frac{1}{N} \sum_{i=1}^{N} e_{\bar{y}_i}^\top (T^{-1}) \ell(\mathbf{g}(x_i)) \tag{4}$$

URE, however, can go below zero during the optimization procedure, leading to overfitting of the model. To address this issue, (Ishida et al., 2019) proposed two tricks, non-negative risk estimator (NN) and gradient accent(GA). The former zeros out the gradient when the mini-batch loss goes below zero while the latter reverse the mini-batch gradient when the loss from any of the complementary class goes below zero. We replace the transition matrix $T$ in the risk estimator 4 with $T_u$ and $T_e$ for **URE-GA-U** and **URE-GA-R**.

## B.2   T-agnostic CLL algorithms

T-agnostic CLL algorithms are those that do not take the information of the transition matrix, includes but not limited to Surrogate complementary loss (SCL) and Discriminative modeling (L-W/L-UW).

**SCL**  (Chou et al., 2020) proposed to use the surrogate complementary loss (SCL) to address the overfitting tendency in URE. The loss function is defined as follows,

$$R(\mathbf{g}) = \frac{1}{N} \sum_{i=1}^{N} \phi(\bar{y}_i, \mathbf{g}(x_i)), \tag{5}$$

where $\phi(\cdot)$ is a surrogate loss for $0 - 1$ loss. For instance, SCL-NL uses the negative log loss $\phi(\bar{y}, \mathbf{g}(\mathbf{x})) = -\log(1 - p_{\bar{y}})$ and SCL-EXP uses the exponential loss $\phi(\bar{y}, \mathbf{g}(\mathbf{x})) = \exp(p_{\bar{y}})$.

**L-W/L-UW**  (Gao & Zhang, 2021) proposed to use discriminative modeling to directly model the distribution of complementary labels. To do so, they proposed the following loss functions,

$$R(\mathbf{g}) = \frac{1}{N} \sum_{i=1}^{N} -\log(\text{softmax}(1 - \text{softmax}(\mathbf{g}(x))))_{\bar{y}_i}, \tag{6}$$

They also proposed a weighting function to further improve the performance. The unweighted version is denoted as L-UW and the weighted version is denoted as L-W.

### B.3 Robust loss methods

(Ishiguro et al., 2022) studied two conditions on loss functions: weighted symmetric condition and relaxation of weighted symmetric condition. Five loss functions that can be robust against the estimation error of the transition matrix were proposed. Their results can be further generalized to noisy complementary label learning. More experiment details for reproduction can be found in their paper.

## C  Additional charts for CLCIFAR dataset with data cleaning

We remove 0%, 25%, 50%, 75%, 100% of the noisy data in CLCIFAR10 and CLCIFAR20 datasets. We discover that by removing the noisy data in the practical dataset, the practical performance gaps vanish for all the CLL algorithms. Therefore, we can conclude that the main obstacle to the practicality of CLL is label noise.

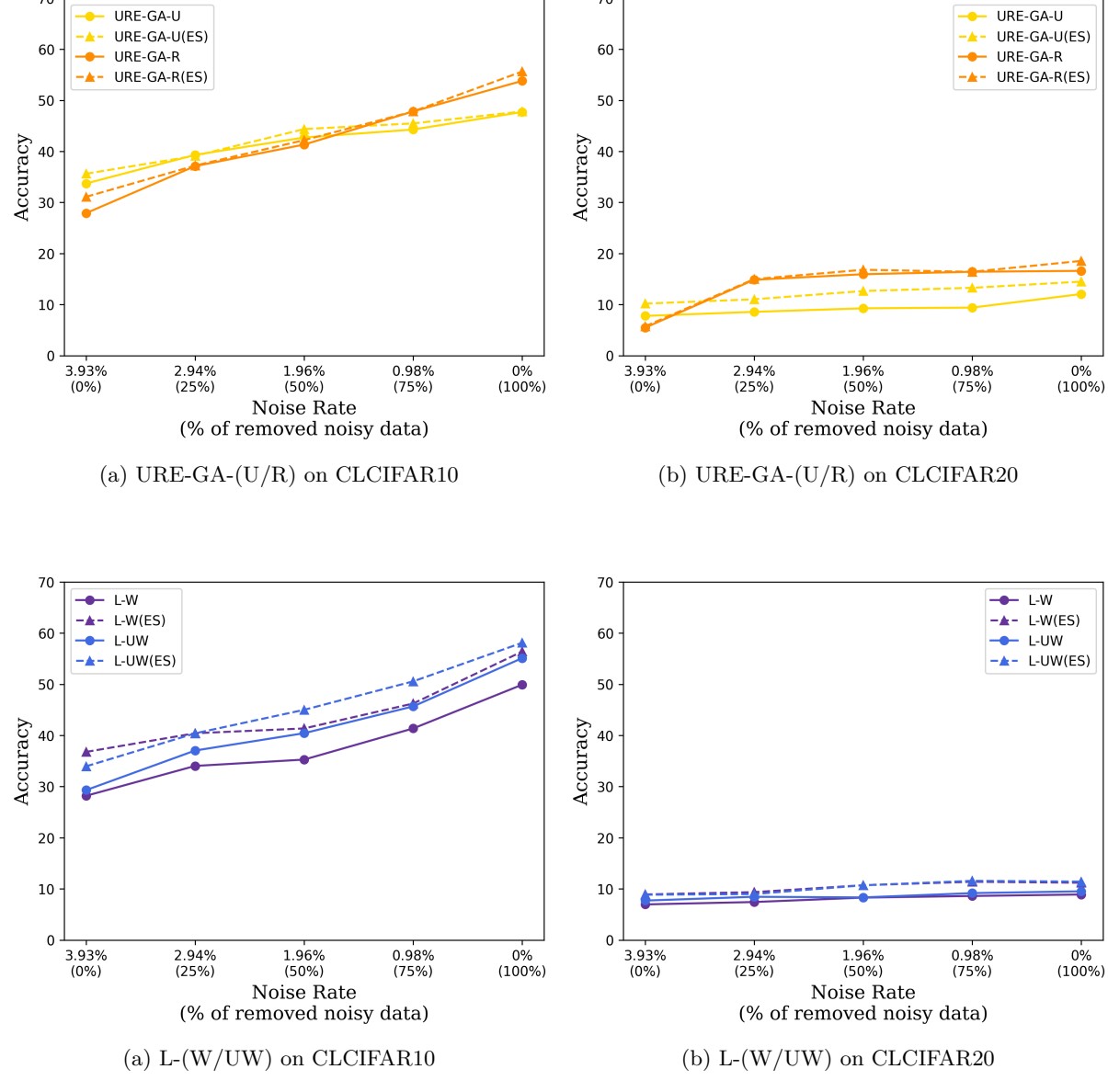

(a) URE-GA-(U/R) on CLCIFAR10

(b) URE-GA-(U/R) on CLCIFAR20

(a) L-(W/UW) on CLCIFAR10

(b) L-(W/UW) on CLCIFAR20

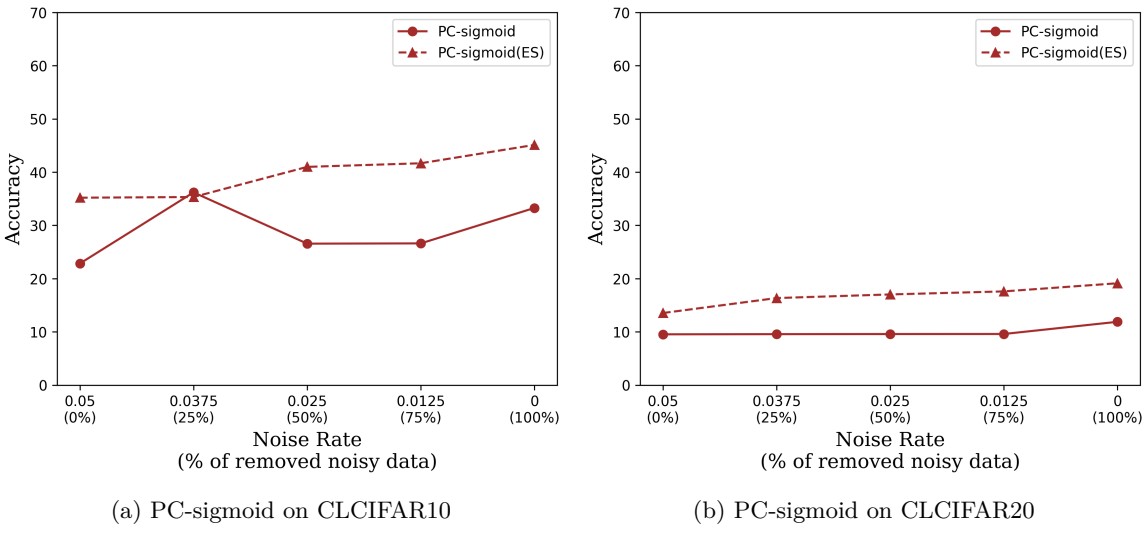

(a) PC-sigmoid on CLCIFAR10

(b) PC-sigmoid on CLCIFAR20

## D   Analysis between multiple label collection trials

We carried out the same protocol for three independent trials to ensure the consistency of our results. The noise rates of CLCIFAR10 are 0.0398, 0.03882, and 0.03928 for three trials respectively. On the other hand, the noise rates of CLCIFAR20 are 0.02322, 0.02902, and 0.03196. These results show that the obtained noise rates are reliable and consistent. Besides, we also analyzed the distribution of complementary label within three trials as reported in Figure 13. The consistent distribution of complementary labels reveals the empirical human annotating biasedness within our protocol. Both analyses show that our protocol and discovery are solid and stable.

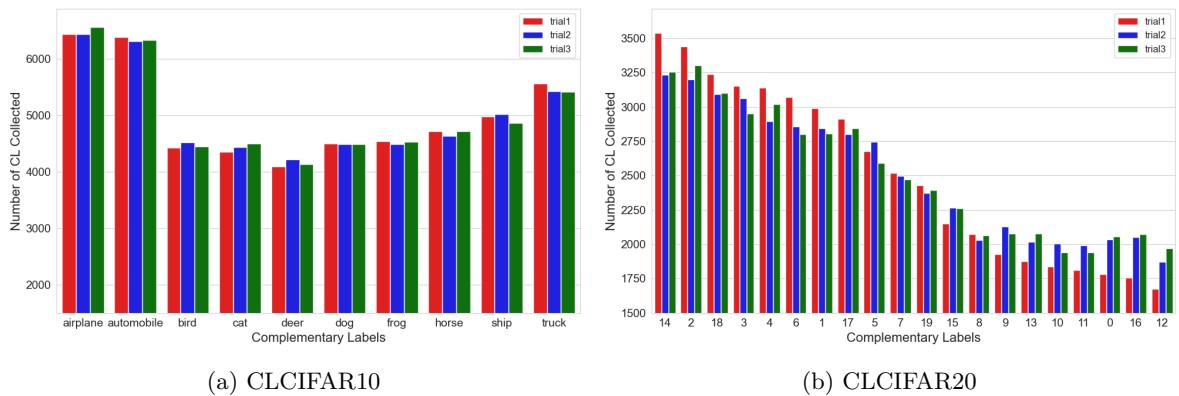

(a) CLCIFAR10

(b) CLCIFAR20

Figure 13: The complementary label distribution of three independent trials of CLCIFAR10 dataset (**Left**) and CLCIFAR20 dataset (**Right**). The full label names of CLCIFAR20 are provided in Appendix E.

# E   Label names of CLCIFAR20 and CLMicroImageNet20

Table 10: The correspondence between index and label names of CLCIFAR20 and CLMicroImageNet20 datasets.

| Index | CLCIFAR20 Label Name | CLMicroImageNet20 Label Name |
|---|---|---|
| 0 | aquatic mammals | tailed frog |
| 1 | fish | scorpion |
| 2 | flowers | snail |
| 3 | food containers | american lobster |
| 4 | fruit, vegetables and mushrooms | tabby |
| 5 | household electrical devices | persian cat |
| 6 | household furniture | gazelle |
| 7 | insects | chimpanzee |
| 8 | large carnivores and bear | bannister |
| 9 | large man-made outdoor things | barrel |
| 10 | large natural outdoor scenes | christmas stocking |
| 11 | large omnivores and herbivores | gasmask |
| 12 | medium-sized mammals | hourglass |
| 13 | non-insect invertebrates | iPod |
| 14 | people | scoreboard |
| 15 | reptiles | snorkel |
| 16 | small mammals | suspension bridge |
| 17 | trees | torch |
| 18 | transportation vehicles | tractor |
| 19 | non-transportation vehicles | triumphal arch |

Table 11: The selected classes/folders for MicroImageNet10 (MIN10) and MicroImageNet20 (MIN20) are drawn from the TinyImageNet200 dataset. The labels provided in the table represent the **first** ordinary label for these classes.

| Index | MIN10 Folder | MIN10 Label Name | Index | MIN20 Folder | MIN20 Label Name |
|---|---|---|---|---|---|
| 0 | n02281406 | sulphur-butterfly | 0 | n01644900 | tailed frog |
| 1 | n02769748 | backpack | 1 | n01770393 | scorpion |
| 2 | n02963159 | cardigan | 2 | n01944390 | snail |
| 3 | n03617480 | kimono | 3 | n01983481 | american lobster |
| 4 | n03706229 | magnetic-compass | 4 | n02123045 | tabby |
| 5 | n03838899 | oboe | 5 | n02123394 | persian cat |
| 6 | n04133789 | scandal | 6 | n02423022 | gazelle |
| 7 | n04456115 | torch | 7 | n02481823 | chimpanzee |
| 8 | n07873807 | pizza | 8 | n02788148 | bannister |
| 9 | n09193705 | alp | 9 | n02795169 | barrel |
| | | | 10 | n03026506 | christmas stocking |
| | | | 11 | n03424325 | gasmask |
| | | | 12 | n03544143 | hourglass |
| | | | 13 | n03584254 | iPod |
| | | | 14 | n04149813 | scoreboard |
| | | | 15 | n04251144 | snorkel |
| | | | 16 | n04366367 | suspension bridge |
| | | | 17 | n04456115 | torch |
| | | | 18 | n04465501 | tractor |
| | | | 19 | n04486054 | triumphal arch |

## F  AutoAugment

In addition to the standard data augmentation, RandomCrop and RandomHorizontalFlip, we also considered a more advanced one, AutoAugment (Cubuk et al., 2019). The benchmark results using AutoAugment are provided in Table 12. We observe that AutoAugment can improve the performance in almost all of the scenarios with a cost of around double running time compared to standard data augmentation. Also, the overfitting tendency of the previous algorithms remains unsolved although we observe that early-stopping can still deliver better performance when using AutoAugment.

Table 12: Comparison of performance using **AutoAugment** on CLCIFAR and uniform-CIFAR datasets in relation to tab:exp-1. The accuracy changes are shown in subscript, with enhanced accuracy values being highlighted in bold.

| methods | uniform-CIFAR10 | | CLCIFAR10 | | uniform-CIFAR20 | | CLCIFAR20 | |
|---|---|---|---|---|---|---|---|---|
| | valid_acc | valid_acc (ES) | valid_acc | valid_acc (ES) | valid_acc | valid_acc (ES) | valid_acc | valid_acc (ES) |
| FWD-U | $\mathbf{75.72}_{+6.55}$ | $\mathbf{77.02}_{+7.23}$ | $\mathbf{42.46}_{+8.37}$ | $\mathbf{43.52}_{+6.69}$ | $\mathbf{26.8}_{+6.56}$ | $\mathbf{26.93}_{+6.31}$ | $7.38_{-0.09}$ | $\mathbf{8.76}_{+0.49}$ |
| FWD-R | $\mathbf{75.53}_{+5.79}$ | $\mathbf{76.06}_{+6.47}$ | $\mathbf{40.37}_{+11.49}$ | $\mathbf{42.13}_{+3.23}$ | $\mathbf{26.74}_{+6.74}$ | $\mathbf{26.98}_{+6.27}$ | $\mathbf{20.71}_{+4.57}$ | $\mathbf{24.77}_{+4.46}$ |
| URE-GA-U | $\mathbf{60.24}_{+5.62}$ | $\mathbf{59.9}_{+4.96}$ | $\mathbf{37.78}_{+3.19}$ | $\mathbf{38.48}_{+2.09}$ | $\mathbf{17.08}_{+1.67}$ | $\mathbf{18.59}_{+2.0}$ | $\mathbf{8.88}_{+1.29}$ | $9.7_{-0.36}$ |
| URE-GA-R | $\mathbf{58.36}_{+5.06}$ | $\mathbf{59.42}_{+2.40}$ | $\mathbf{31.98}_{+3.28}$ | $\mathbf{33.08}_{+2.14}$ | $\mathbf{18.2}_{+3.34}$ | $\mathbf{19.72}_{+2.39}$ | $\mathbf{10.85}_{+5.61}$ | $\mathbf{9.89}_{+4.43}$ |
| SCL-NL | $\mathbf{76.6}_{+9.45}$ | $\mathbf{76.83}_{+8.19}$ | $\mathbf{38.4}_{+4.6}$ | $\mathbf{43.22}_{+5.41}$ | $\mathbf{23.11}_{+3.07}$ | $\mathbf{26.62}_{+5.94}$ | $7.34_{-0.24}$ | $8.34_{-0.19}$ |
| SCL-EXP | $\mathbf{75.9}_{+11.04}$ | $\mathbf{75.75}_{+10.35}$ | $\mathbf{40.95}_{+6.36}$ | $\mathbf{41.63}_{+4.67}$ | $\mathbf{24.96}_{+5.56}$ | $\mathbf{26.64}_{+5.61}$ | $7.21_{-0.34}$ | $\mathbf{8.47}_{+0.36}$ |
| L-W | $\mathbf{67.2}_{+10.99}$ | $\mathbf{71.07}_{+11.89}$ | $\mathbf{33.89}_{+5.85}$ | $\mathbf{38.16}_{+3.61}$ | $\mathbf{22.28}_{+7.93}$ | $\mathbf{23.19}_{+4.08}$ | $\mathbf{7.58}_{+0.5}$ | $8.64_{-0.1}$ |
| L-UW | $\mathbf{72.39}_{+11.51}$ | $\mathbf{73.26}_{+10.83}$ | $\mathbf{34.61}_{+3.98}$ | $\mathbf{40.3}_{+5.17}$ | $\mathbf{23.31}_{+7.3}$ | $\mathbf{24.41}_{+4.99}$ | $\mathbf{7.47}_{+0.11}$ | $\mathbf{8.96}_{+0.25}$ |
| PC-sigmoid | $\mathbf{45.72}_{+17.52}$ | $\mathbf{46.53}_{+7.24}$ | $\mathbf{33.24}_{+8.86}$ | $\mathbf{40.72}_{+4.84}$ | $\mathbf{12.81}_{+3.09}$ | $13.84_{-2.61}$ | $\mathbf{14.15}_{+4.88}$ | $\mathbf{17.06}_{+2.8}$ |
| MAE | $\mathbf{61.26}_{+3.89}$ | $\mathbf{63.41}_{+4.91}$ | $\mathbf{21.74}_{+5.44}$ | $\mathbf{23.65}_{+4.21}$ | $\mathbf{20.03}_{+3.41}$ | $\mathbf{21.79}_{+4.16}$ | $\mathbf{5.18}_{+0.07}$ | $\mathbf{6.68}_{+0.81}$ |

## G  Datasheets for Datasets

We refer to the template and guiding questions provided in 'Datasheets for Datasets' by (Gebru et al., 2021).

### G.1  Motivation

**For what purpose was the dataset created?** Was there a specific task in mind? *Was there a specific gap that needed to be filled? Please provide a description.*

The dataset was created to enable research on complementary label learning to gain insights into the real-world performance of CLL algorithms, we developed a protocol to collect complementary labels from human annotators.

**Who created the dataset (e.g., which team, research group) and on behalf of which entity (e.g., company, institution, organization)?**

The datasets were created by: Computational Learning Lab, Dept. of Computer Science and Information Engineering, National Taiwan University.

**Who funded the creation of the dataset?** *If there is an associated grant, please provide the name of the grantor and the grant name and number.*

Funding was provided from those distinct sources: the National Science and Technology Council in Taiwan via NSTC 113-2628-E-002-003, NSTC 113-2634-F-002-008, NSTC 112-2628-E-002-030, and National Center for High-performance Computing (NCHC) of National Applied Research Laboratories (NARLabs) in Taiwan for providing computational and storage resources.

**Any other comments?**

None.

### G.2  Composition

**What do the instances that comprise the dataset represent (e.g., documents, photos, people, countries)?** *Are there multiple types of instances (e.g., movies, users, and ratings; people and interactions between them; nodes and edges)? Please provide a description.*

The instances are the images which are derived the published datasets including CIFAR10, CIFAR100, TinyImageNet.

**How many instances are there in total (of each type, if appropriate)?**

The CLCIFAR10 and CLCIFAR20 datasets each contain 50,000 training instances and 10,000 testing instances. For the CLMicroImageNet datasets, CLMicroImageNet10 has 5,000 training instances and 500 testing instances, whereas CLMicroImageNet20 includes 10,000 training instances and 1,000 testing instances.

**Does the dataset contain all possible instances or is it a sample (not necessarily random) of instances from a larger set?** *If the dataset is a sample, then what is the larger set? Is the sample representative of the larger set (e.g., geographic coverage)? If so, please describe how this representativeness was validated/verified. If it is not representative of the larger set, please describe why not (e.g., to cover a more diverse range of instances, because instances were withheld or unavailable).*

For details regarding the CLCIFAR10 and CLCIFAR20 datasets, please refer to Section 3.1. The CLMicroImageNet10 and CLMicroImageNet20 datasets are described in Appendix A.5.

**What data does each instance consist of?** *"Raw" data (e.g., unprocessed text or images)or features? In either case, please provide a description.*

Each instance consists of an image associated with three complementary labels. The images in the CLCIFAR10 and CLCIFAR20 datasets have dimensions of 32x32 pixels, while those in CLMicroImageNet10 and CLMicroImageNet20 have dimensions of 64x64 pixels.

**Is there a label or target associated with each instance?** *If so, please provide a description.*

The label is the negative labels of picture, called complementary label. The complementary label derived from other labels, which an instance does not belong.

**Is any information missing from individual instances?** *If so, please provide a description, explaining why this information is missing (e.g., because it was unavailable). This does not include intentionally removed information, but might include, e.g., redacted text.*

Everything is included. No data is missing.

**Are relationships between individual instances made explicit (e.g.,users' movie ratings, social network links)?** *If so, please describe how these relationships are made explicit.*

None explicitly.

**Are there recommended data splits (e.g., training, development/validation, testing)?** *If so, please provide a description of these splits, explaining the rationale behind them.*

The datasets are included the testing set. Results are measured in classification accuracy.

**Are there any errors, sources of noise, or redundancies in the dataset?** *If so, please provide a description.*

Please refer to Section 4, and Appendix A.4 for detailed information.

**Is the dataset self-contained, or does it link to or otherwise rely on external resources (e.g., websites, tweets, other datasets)?** *If it links to or relies on external resources, a) are there guarantees that they will exist, and remain constant, over time; b) are there official archival versions of the complete dataset (i.e., including the external resources as they existed at the time the dataset was created); c) are there any restrictions (e.g., licenses, fees) associated with any of the external resources that might apply to a dataset*

*consumer? Please provide descriptions of all external resources and any restrictions associated with them, as well as links or other access points, as appropriate.*

The dataset is entirely self-contained.

**Does the dataset contain data that might be considered confidential (e.g., data that is protected by legal privilege or by doctor–patient confidentiality, data that includes the content of individuals' nonpublic communications)?** *If so, please provide a description.*

All are image which contents vehicles, animals, etc. It is not considered confidential.

**Does the dataset contain data that, if viewed directly, might be offensive, insulting, threatening, or might otherwise cause anxiety?** *If so, please describe why.*

No.

**Does the dataset identify any subpopulations (e.g., by age, gender)?** *If so, please describe how these subpopulations are identified and provide a description of their respective distributions within the dataset.*

No.

**Is it possible to identify individuals (i.e., one or more natural persons), either directly or indirectly (i.e., in combination with other data) from the dataset?** *If so, please describe how.*

No.

**Does the dataset contain data that might be considered sensitive in any way (e.g., data that reveals race or ethnic origins, sexual orientations, religious beliefs, political opinions or union memberships, or locations; financial or health data; biometric or genetic data; forms of government identification, such as social security numbers; criminal history)?** *If so, please provide a description.*

No.

**Any other comments?**

None.

### G.3    Collection Process

**How was the data associated with each instance acquired?** Was the data directly observable (e.g., raw text, movie ratings), reported by subjects (e.g., survey responses), or indirectly inferred/derived from other data (e.g., part-of-speech tags, model-based guesses for age or language)? *If the data was reported by subjects or indirectly inferred/derived from other data, was the data validated/verified? If so, please describe how.*

The data was mostly observable as raw image. The data was collected by downloading the link at: `https://www.cs.toronto.edu/~kriz/cifar.html`, `http://cs231n.stanford.edu/tiny-imagenet-200.zip`.

**What mechanisms or procedures were used to collect the data (e.g., hardware apparatuses or sensors, manual human curation, software programs, software APIs)?** *How were these mechanisms or procedures validated?*

Please refer to Section 3.2 for detailed information.

**If the dataset is a sample from a larger set, what was the sampling strategy (e.g., deterministic, probabilistic with specific sampling probabilities)?**

Please refer to Section 3.1, and Appendix A.5 for detailed information.

**Who was involved in the data collection process (e.g., students, crowdworkers, contractors) and how were they compensated (e.g., how much were crowdworkers paid)?**

The labeling tasks were deployed on Amazon MTurk by dividing the overall dataset into manageable Human Intelligence Tasks (HITs). Specifically, to construct the CLCIFAR and CLMicroImageNet datasets, we first partitioned the total of 50,000 images into five batches, each containing 10,000 images. Each batch was then further divided into 1,000 HITs, with each HIT containing 10 images. Each HIT was assigned to three annotators, who received a reward of 0.03 dollar per HIT (10 images). The decision to set the reward at 0.03 dollar per HIT was based on the understanding that assigning complementary labels requires less effort and lower expertise compared to assigning true labels. We conducted preliminary research on typical reward ranges on Amazon MTurk and determined that this amount appropriately compensates annotators at approximately 6 dollar per hour, reflecting a fair balance between annotator effort, data quality, and cost-effectiveness.

**Over what timeframe was the data collected?** Does this timeframe match the creation timeframe of the data associated with the instances (e.g., recent crawl of old news articles)? *If not, please describe the timeframe in which the data associated with the instances was created.*

The CLCIFAR datasets have been collected in 2023 and CLMicroImageNet has been collected in 2024 to expand our complementary datasets.

**Were any ethical review processes conducted (e.g., by an institutional review board)?** *If so, please provide a description of these review processes, including the outcomes, as well as a link or other access point to any supporting documentation.*

N/A.

**Did you collect the data from the individuals in question directly, or obtain it via third parties or other sources (e.g., websites)?**

All datasets are labeled by human annotators on Amazon Mechanical Turk (MTurk).

**Were the individuals in question notified about the data collection?** *If so, please describe (or show with screenshots or other information) how notice was provided, and provide a link or other access point to, or otherwise reproduce, the exact language of the notification itself.*

No. All data have been asked the same question "What is the incorrect label of the picture?" and following by four multiple choices.

**Did the individuals in question consent to the collection and use of their data?** *If so, please describe (or show with screenshots or other information) how consent was requested and provided, and provide a link or other access point to, or otherwise reproduce, the exact language to which the individuals consented.*

N/A.

**If consent was obtained, were the consenting individuals provided with a mechanism to revoke their consent in the future or for certain uses?** *If so, please provide a description, as well as a link or other access point to the mechanism (if appropriate).*

N/A.

**Has an analysis of the potential impact of the dataset and its use on data subjects (e.g., a data protection impact analysis) been conducted?** *If so, please provide a description of this analysis, including the outcomes, as well as a link or other access point to any supporting documentation.*

N/A.

**Any other comments?**

None.

### G.4 Preprocessing/cleaning/labeling

**Was any preprocessing/cleaning/labeling of the data done (e.g., discretization or bucketing, tokenization, part-of-speech tagging, SIFT feature extraction, removal of instances, processing**

**of missing values)?** *If so, please provide a description. If not, you may skip the remaining questions in this section.*

None.

**Was the "raw" data saved in addition to the preprocessed/cleaned/labeled data (e.g., to support unanticipated future uses)?** *If so, please provide a link or other access point to the "raw" data.*

**Is the software that was used to preprocess/clean/label the data available?** *If so, please provide a link or other access point.*

**Any other comments?**

None.

### G.5   Uses

**Has the dataset been used for any tasks already?** *If so, please provide a description.*

This datasets have been derived from public datasets–CIFAR10, CIFAR100, TinyImageNet200– which are popular datasets from classification learning in Computer Vision domain.

**Is there a repository that links to any or all papers or systems that use the dataset?** *If so, please provide a link or other access point.*

For detail information, please refer our repository.

**What (other) tasks could the dataset be used for?**

The dataset could be used for anything related to modeling or understanding of how to model learning from inexact, incorrect data (complementary label learning).

**Is there anything about the composition of the dataset or the way it was collected and preprocessed/cleaned/labeled that might impact future uses?** *For example, is there anything that a dataset consumer might need to know to avoid uses that could result in unfair treatment of individuals or groups (e.g., stereotyping, quality of service issues) or other risks or harms (e.g., legal risks, financial harms)? If so, please provide a description. Is there anything a dataset consumer could do to mitigate these risks or harms?*

None.

**Are there tasks for which the dataset should not be used?** *If so, please provide a description.*

The datasets are collected specifically for complementary label learning, where models are trained using weaker labeling information rather than exact true labels. Consequently, predictions about true labels are inferred indirectly. Due to this indirect labeling approach, there is a potential increased risk of privacy leakage if applied to sensitive or personally identifiable image data. Thus, the datasets should not be used in contexts where the protection of user privacy is critical, or where indirect inferences might inadvertently compromise sensitive personal information.

**Any other comments?**

None.

### G.6   Distribution

**Will the dataset be distributed to third parties outside of the entity (e.g., company, institution, organization) on behalf of which the dataset was created?** *If so, please provide a description.*

Yes, the dataset is publicly available on the internet.

**How will the dataset will be distributed (e.g., tarball on website, API, GitHub)?** *Does the dataset have a digital object identifier (DOI)?*

The dataset will be hosted on our GitHub to ensure accessibility to the community. For the official DOI, please refer the official publication on TMLR.

**When will the dataset be distributed?**

The original datasets were first released in 2012 for CIFAR and 2015 for TinyImageNet. Our complementary dataset is released via this link.

**Will the dataset be distributed under a copyright or other intellectual property (IP) license, and/or under applicable terms of use (ToU)?** *If so, please describe this license and/or ToU, and provide a link or other access point to, or otherwise reproduce, any relevant licensing terms or ToU, as well as any fees associated with these restrictions.*

The crawled data copyright belongs to the authors of the reviews unless otherwise stated. There is no license, but there is a request to cite the corresponding paper if the dataset is used: *Learning multiple layers of features from tiny images*, Alex Krizhevsky, University of Toronto, 05 2012, and *Tiny imagenet visual recognition challenge*, Ya Le and Xuan S. Yang, 2015. The CLImage datasets and accompanying code are available under the *MIT License*. For further information, please refer our GitHub.

**Have any third parties imposed IP-based or other restrictions on the data associated with the instances?** *If so, please describe these restrictions, and provide a link or other access point to, or otherwise reproduce, any relevant licensing terms, as well as any fees associated with these restrictions.*

No.

**Do any export controls or other regulatory restrictions apply to the dataset or to individual instances?** *If so, please describe these restrictions, and provide a link or other access point to, or otherwise reproduce, any supporting documentation*

These datasets have been driven from CIFAR10, CIFAR100, TinyImageNet200, which are known as the public datasets.

**Any other comments?**

None.

### G.7 Maintenance

**Who will be supporting/hosting/maintaining the dataset?**

Computational Learning Lab, Dept. of Computer Science and Information Engineering, National Taiwan University.

**How can the owner/curator/manager of the dataset be contacted (e.g., email address)?**

Prof. Hsuan-Tien Lin, `htlin@csie.ntu.edu.tw`.

**Is there an erratum?** *If so, please provide a link or other access point.*

Currently, this dataset just has one version.

**Will the dataset be updated (e.g., to correct labeling errors, add new instances, delete instances)?** If so, please describe how often, by whom, and how updates will be communicated to dataset consumers (e.g., mailing list, GitHub)?

All updates and changes related to the CLImage dataset will be reflected in this repository.

**If the dataset relates to people, are there applicable limits on the retention of the data associated with the instances (e.g., were the individuals in question told that their data would be retained for a fixed period of time and then deleted)?** If so, please describe these limits and explain how they will be enforced.

N/A.

**Will older versions of the dataset continue to be supported/hosted/maintained?** *If so, please describe how. If not, please describe how its obsolescence will be communicated to dataset consumers.*

This dataset has been hosted and maintained via our repository.

**If others want to extend/augment/build on/contribute to the dataset, is there a mechanism for them to do so?** *If so, please provide a description. Will these contributions be validated/verified? If so, please describe how. If not, why not? Is there a process for communicating/distributing these contributions to dataset consumers? If so, please provide a description.*

Others may do so and should contact the original authors about incorporating fixes/extensions.

**Any other comments?**

None.

