# OpenReview forum: "CLImage: Human-Annotated Datasets for Complementary-Label Learning"
_TMLR — Accepted by TMLR_

### Review · Reviewer_1Ytp · 2025-03-25

**Summary Of Contributions:**

The task the paper focuses on is the Complementary-label learning (CLL). While the previous methods on CLL were built upon the synthetic dataset with certain assumptions about the generation of complementary labels, the authors claim that this assumption is far from the human behavior, brining a distribution gap. The authors develop a protocol to collect complementary labels from human annotators based on CIFAR10, CIFAR100, and TinyImageNet200, suggesting 4 datasets: CLCIFAR10, CLCIFAR20, CLMicroImageNet10, CLMicroImageNet20. The study reveals that annotation noise is the most influential factor, and the gap in the annotation distribution between the synthetic dataset and their human-labeled dataset is attributed to a huge performance gap when using state-of-the-art methods for CLL.

**Audience:**

Yes

**Broader Impact Concerns:**

The broader impact is somewhat described.

**Claims And Evidence:**

Yes

**Requested Changes:**

I would like to hear about the authors' thoughts on the weaknesses mentioned above.

**Strengths And Weaknesses:**

Strengths:
- The paper is well-written, and it's easy to follow.
- The motivation for the need for human-annotated datasets for CLL is well described.
- 4 datasets are suggested to show the consistency of their protocol in building the CLL dataset.
- Enough ablation studies have been conducted.

Weaknesses:
- When annotating the image, the authors asked annotators to select a "wrong" label. This is why there's a biased transition matrix with irrelevant classes between true labels and complementary labels in Figure 2. What I think is more natural is to ask annotators to select "the best" label among the given four labels without including the true label. The reason I said natural is that people tend to wrongly label when it's difficult to distinguish between true class and wrongly annotated class (e.g. "river" instead of "pond" when the image is about a satellite pond image). This thought is somewhat similar to [A] which suggests image-dependent noisy annotation in the partial label learning.

[A] Instance-Dependent Multilabel Noise Generation for Multilabel Remote Sensing Image Classification, IEEE JSTARS.

---

> ### Author Response · Authors · 2025-04-14
>
> Thanks for your question. This is a very good point.
>
> Following your suggestion, we added a new section “**Alternative to Current Data Collection Protocol**”. In this section, we discussed several protocols, including the protocol that you suggested, and the key results of additional analysis where the true label was not included among the provided labels is very similar to the result of our protocol. We also observed a persistent biased transition matrix, as illustrated in Figure 7 of our revised version.
>
> We hope that this explanation will inspire future work on other possible collection protocols of complementary labels.

---

### Review · Reviewer_eRfs · 2025-03-30

**Summary Of Contributions:**

The authors of this paper address the problem of **complementary label learning** (CLL), where a set of complementary labels is assigned per instance. In contrast to ordinary multi-class classification, such a set indicates the classes to which the respective instance does not belong. The authors of this paper identify a lack of real-world datasets with complementary labels to evaluate the practical applicability of CLL approaches. Hence, providing such human-annotated datasets would be a major advancement compared to relying on datasets with synthetic (simulated) complementary labels. For this purpose, the authors make the following key **contributions**:

- The author collect and publish (in the case of acceptance) four datasets, named CLCIFAR10, CLCIFAR20, CLMicroImageNet10, and
CLMicroImageNet20 as CLL variants of well-known image benchmark datasets.
- The authors present a data collection protocol for creating future datasets with complementary labels.
- The authors demonstrate in an extensive empirical analysis the shortcomings of existing CLL approaches and identify concrete reasons for these shortcomings by comparing the performance gap between their real-world CLL datasets with the commonly used synthetic ones.

**In summary**, these contributions lay out the foundations for future advancements and more realistic empirical evaluations in the research area of CLL.

**Audience:**

Yes

**Broader Impact Concerns:**

Complementary labels were collected using crowdworkers on Amazon Mechanical Turk. It is essential to ensure that these workers receive fair compensation. Therefore, in addition to stating the payment provided, a detailed explanation of the rationale behind the payment structure would be valuable. Such a rationale will be also required when including the aforementioned datasheet.

**Claims And Evidence:**

Yes

**Requested Changes:**

**Requested Changes:** I would appreciate it if the authors could address the aforementioned *weaknesses* and clarify the following *questions*, *minor remarks*, and the *broader impact concerns* raised below. If I have misunderstood or overlooked any aspect, I am willing to reconsider my perspective on these issues. Looking forward to a constructive discussion!

**Questions:**
- Will anonymized identifiers of the crowdworkers / annotators be published? This could be helpful to understand the worker-dependent component (e.g., different noise levels) when acquiring complementary labels.
- Does user refer to the identity of the crowdworkers / annotators in the following statement as part of your *Broader Impact Statement*:
> The privacy of the users may be easier to compromised as a result.
- Will the code for labeling interface on Amazon Mechanical Turk be published to ease future crowdworking studies?

**Minor Remarks**:
- In my opinion, the appendices must not be part of the supplementary material but can follow after the references in the main paper. This will ease the access to the appendices.
- Section 1:
> Moreover, through ~thorough~ analysis we confirmed that ...
- I would propose to use \cite instead of \citep when using the reference as a direct part of the sentence, such that the authors names are not in round brackets. For example, this would correspond to the following change in Section 2.3:
> The pioneering work by ~(Ishida et al., 2017)~ Ishida et al. (2017) studied
- Appendix A.1:
> ... and described the process how MicroImageNet10 and MicroImageNet20 datasets ~generation~ were generated ...
- Figure 1 and 7 contain graphs with ticks of rather small font sizes on their x-axes. These should be increase, if possible. Moreover, when zooming in, the font is quite blurry. Maybe you can use PDFs to avoid such things. Feel free to check these comments also for the other figures to improve the overall presentation of your paper.
- The notation of the *softmax* function is inconsistent when comparing Eq. (2) in the main paper with Eq. (3) in the appendices.
- By changing the column order in Table 1, a direct comparison between uniform-CIFAR10 and CLCIFAR10 will be easier.

**Strengths And Weaknesses:**

Below, I list the *strengths* and *weaknesses* according to their importance, starting with the most important one and ending with the least important one.

**Strengths:**
- At the core of this paper, the authors contribute four *datasets with real-world complementary labels*, which is a key requirement for a realistic evaluation of CLL approaches. Moreover, the authors do not only restrict this contribution to rather low complex images of the CIFAR datasets but also include more complex images of the TinyImageNet dataset. Further, different numbers of classes as an important factor in CLL are covered.
- The realistic *evaluation of CLL approaches* on these novel datasets is well-structured. This means that instead of only reporting the results for these new datasets, the authors additionally compare these results to the existing procedure of simulating complementary labels and identify the label noise of the crowdworkers as major reasons for the performance drops of the CLL approaches.
- A simple *data collection protocol* allows an easy extension of this data collection by performing similar crowdworking studies for new datasets.
- The writing and presentation is mostly clear throughout the paper. As a result, and together with the extended literature overview of CLL approaches in the appendix as supplementary material, readers can easily understand the major takeaways of this paper.

**Weaknesses**:
- Since the publication of the collected complementary labels corresponds to new variants of existing benchmark datasets, I would expect a datasheet [1] detailing the dataset's documentation and usage. Such a sheet has also been recommended by data-centric venues such as the Journal on *Data-centric Machine Learning Research* (DMLR) [2] and the *NeurIPS: Datasets and Benchmarks Tracks* in 2024 [3].
- The hosting platform of the data and the publication of the associated code for the evaluation is unclear. I'm aware that there is no link due to the double-blind requirement, yet brief information regarding these two aspects will be helpful for my assessment as a reviewer. Moreover, the code available during review, e.g., as supplementary material, would be helpful to assess its reproducibility and its future usage benefit for other researchers.
- The labeling evaluation protocol needs further motivation and reasoning. For example, the specification of the concrete number of four potential classes in the prefiltering step for selecting the complementary labels is unclear. This is in particular questionable for numerous classes.
- I understand that ResNet18 is still a common architecture for training on the CIFAR datasets and subsets of TinyImageNet. Yet, readers could question the choice of this architecture and also expect results with more modern neural network architectures.

**References**:
- [1] Gebru, Timnit, Jamie Morgenstern, Briana Vecchione, Jennifer Wortman Vaughan, Hanna Wallach, Hal Daumé Iii, and Kate Crawford. "Datasheets for datasets." Communications of the ACM 64, no. 12 (2021): 86-92.
- [2] DMLR: https://data.mlr.press/submissions.html
- [3] NeurIPS: https://neurips.cc/Conferences/2024/CallForDatasetsBenchmarks

---

> ### Author Response · Authors · 2025-04-14
>
> Thank you for your valuable questions and suggestions. Please find our answer below:
>
> **Question 1**: Since the publication of the collected complementary labels corresponds to new variants of existing benchmark datasets, I would expect a datasheet [1] detailing the dataset's documentation and usage. Such a sheet has also been recommended by data-centric venues such as the Journal on Data-centric Machine Learning Research (DMLR) [2] and the NeurIPS: Datasets and Benchmarks Tracks in 2024 [3].
>
> **Answer 1**: Thank you for your valuable comment. We have included “**Datasheets of Datasets**” in **Appendix H** of the paper that followed the datasheets template of NeurIPS. There are some questions that we will update with detailed information later, after the review process has been done to adhere to the double-blink review.
> We hope these datasheets will fulfill and cover all concerns relevant to datasets. We appreciate your valuable feedback to provide greater clarity and enrich the manuscript.
>
> **Question 2**: The hosting platform of the data and the publication of the associated code for the evaluation is unclear. I'm aware that there is no link due to the double-blind requirement, yet brief information regarding these two aspects will be helpful for my assessment as a reviewer. Moreover, the code available during review, e.g., as supplementary material, would be helpful to assess its reproducibility and its future usage benefit for other researchers.
>
> **Answer 2**: Thanks for your reasonable suggestion.
> We included all source codes, datasets, and temporary shares in an anonymous repository “*https://anonymous.4open.science/r/CLImage_Dataset-5CFB*”. We also added a section in **Appendix E** to clarify it. We plan to release all source codes and datasets publicly after completing the review process.
>
> **Question 3**: The labeling evaluation protocol needs further motivation and reasoning. For example, the specification of the concrete number of four potential classes in the prefiltering step for selecting the complementary labels is unclear. This is in particular questionable for numerous classes.
>
> **Answer 3**: Thank you so much for your thoughtful question. In this work, we used a specific protocol to collect complementary labels, but we acknowledge there are several alternative protocols that could also be considered. Following your suggestion, we added a new Section to further explain that in the paper. The name of this section is “**Alternative to Current Data Collection Protocol**” which highlights possibilities while explaining the rationale behind our current design choices. Once again, thanks for your valuable question.
>
> **Question 4**: I understand that ResNet18 is still a common architecture for training on the CIFAR datasets and subsets of TinyImageNet. Yet, readers could question the choice of this architecture and also expect results with more modern neural network architectures.
>
> **Answer 4**: Thank you for this great suggestion. We chose to begin benchmarking with simpler models due to the known issue of overfitting in CLL [1]. Simpler models are typically less prone to overfitting, making them suitable for initial benchmarking. We conducted an ablation study comparing different neural network architectures, including ResNet18, ResNet34, ResNet50, and DenseNet12, to determine the most suitable architecture for our complementary dataset. Results shown in Table 9 in Appendix A.6 indicate that ResNet18 achieved the best performance. Additionally, ResNet18 closely aligns with ResNet architectures commonly used in CLL studies [2],[3]. Thus, we selected ResNet18 for benchmarking purposes.
>
> *references*:
>
> [1] Yu-Ting Chou, Gang Niu, Hsuan-Tien Lin, and Masashi Sugiyama. Unbiased risk estimators can mislead: A case study of learning with complementary labels. In Proceedings of the 37th ICML, pages 1929–1938, 2020.
>
> [2] Nai-Xuan Ye, Tan-Ha Mai, Hsiu-Hsuan Wang, Wei-I Lin, and Hsuan-Tien Lin. libcll: an extendable python toolkit for complementary-label learning, 2024.
>
> [3] Yanwu Xu, Mingming Gong, Junxiang Chen, Tongliang Liu, Kun Zhang, and Kayhan Batmanghelich.
> Generative-discriminative complementary learning, 2019.

---

> > ### Author Response · Authors · 2025-04-14
> >
> > **Question 5**: Will anonymized identifiers of the crowdworkers / annotators be published? This could be helpful to understand the worker-dependent component (e.g., different noise levels) when acquiring complementary labels.
> >
> > **Answer 5**: We agreed with you that publishing that information could be useful to understand the worker-dependent component. In the datasets release, we published the anonymized worker IDs by hashing the original IDs that were provided by the Amazon mechanical terms. We believe this mechanism is efficient in preserving the privacy of the workers. The worker IDs are included in the source code, which is temporarily on anonymous Github “*https://anonymous.4open.science/r/CLImage_Dataset-5CFB*” or the temporary link can be found in **Appendix E**.
> >
> > **Question 6**: Does user refer to the identity of the crowdworkers / annotators in the following statement as part of your Broader Impact Statement: “The privacy of the users may be easier to compromised as a result”?
> >
> > **Answer 6**: Thank you for your question. The privacy concern we mentioned does not refer to the identity of the crowdworkers. Rather, our intended message is that the provided datasets could be used to advance complementary label learning algorithms that infer true labels from limited or ambiguous information. This process may unintentionally expose sensitive or personal details embedded within the data. To clarify this and avoid confusion for readers, we have revised the “Broader Impact Statement” in our updated version as follows:
> > “*It is known that weakly-supervised learning can be used for some privacy-preserving applications. Our CLImage datasets do not belong to those. We suggest practitioners exercise caution when studying such applications.*”
> >
> > **Question 7**: Will the code for labeling interface on Amazon Mechanical Turk be published to ease future crowdworking studies?
> >
> > **Answer 7**:  Thank you for your comment. We shared the code of labeling interface on Amazon Mechanical Turk, which can be found in **Appendix E** or the temporary link “*https://anonymous.4open.science/r/CLImage_Dataset-5CFB*”.
> >
> > **Minor remarks**: Thanks for all your comments. We have solved all the minor remarks, improved all figures, and updated it in the revised version
> >
> > **Broader Impact Concerns**: Thanks for your suggestion. We have included this information in "**Datasheets for Datasets**".

---

> > > ### Comment · Reviewer_eRfs · 2025-04-21
> > >
> > > Many thanks for your detailed answer and the made changes, which address my major concerns.
> > >
> > > Since you mention the annotation effort in the added *Section 7: Alternative to Current Data Collection Protocol*, consider **publishing annotation times** as well. This will give researchers not only the information who provided complementary labels but also how long each labeling of an image took, offering a useful baseline for designing alternative labeling protocols. Ensure your data structure in your repository’s README includes both the annotation time and the annotator ID.
> > >
> > > As another important remark, please **revise the references**, which are currently inconsistent. For example, some works are cited as arXiv despite having formal publications, and venue names are either missing or variably formatted (lowercase, CamelCase, or abbreviations).
> > >
> > > Finally, please **polish your article** by eliminating typos and grammatical errors and ensuring figures and tables are clear and well‐captioned. For instance, revise the caption of Fig. 7 for correct grammar.

---

> > > > ### Author Response · Authors · 2025-04-22
> > > >
> > > > Thank you so much for your suggestion. Please find our response below.
> > > >
> > > > Since you mention the annotation effort in the added Section 7: Alternative to Current Data Collection Protocol, consider **publishing annotation times** as well. This will give researchers not only the information who provided complementary labels but also how long each labeling of an image took, offering a useful baseline for designing alternative labeling protocols. Ensure your data structure in your repository’s README includes both the annotation time and the annotator ID.
> > > >
> > > > --> Thank you for your suggestion. We have prepared and uploaded the annotation times for **CLMIN10** to our anonymous GitHub repository. Retrieving historical annotation times for the other datasets will require additional effort, so we plan to add those once the repository is publicly released. We have also updated the **README** to reflect the inclusion of annotation times.
> > > >
> > > > As another important remark, please **revise the references**, which are currently inconsistent. For example, some works are cited as arXiv despite having formal publications, and venue names are either missing or variably formatted (lowercase, CamelCase, or abbreviations).
> > > >
> > > > --> We have reviewed, updated, and corrected all references to ensure consistency with formal publication formats and venue names. Thank you for your guidance.
> > > >
> > > > Finally, please **polish your article** by eliminating typos and grammatical errors and ensuring figures and tables are clear and well‐captioned. For instance, revise the caption of Fig. 7 for correct grammar.
> > > >
> > > > --> We have reviewed and updated the caption of Figure 7, corrected any typos, and standardized the titles across all figures. We will continue polishing the manuscript to eliminate any remaining typos and grammatical errors
> > > >
> > > > Additionally, we have resubmitted our revised manuscript incorporating your suggestions. Thank you so much!

---

> > > > > ### Comment · Reviewer_eRfs · 2025-05-03
> > > > >
> > > > > Dear authors,
> > > > >
> > > > > many thanks for your made changes further improving your paper.
> > > > >
> > > > > As a **very minor remark**, I'd like to point out that the references can still be polished. For example, the arXiv version of the "Dinov2: Learning robust visual features without supervision" paper is only referenced instead of the one published at TMLR. A similar observation applies to "AutoAugment: Learning Augmentation Strategies from Data", which is published at CVPR. Another example is the inconsistent capitalization of the venue names when comparing names such as "International conference on machine learning" and "International Conference on Artificial Intelligence and Statistics".

---

> > > > > > ### Author Response · Authors · 2025-05-04
> > > > > >
> > > > > > Dear Reviewer **eRfs**,
> > > > > >
> > > > > > Thank you very much for your helpful feedback, which improved the overall quality and clarity of our paper.
> > > > > > In response to your suggestions, we have thoroughly reviewed all references and resolved inconsistencies in capitalization. Additionally, we have updated the citations of papers that were previously listed as arXiv versions to their corresponding official conference or journal publications, where available.
> > > > > > The revised version of the manuscript has been resubmitted to reflect these updates.
> > > > > >
> > > > > > Best regards,

---

### Review · Reviewer_rAqJ · 2025-04-05

**Summary Of Contributions:**

The paper proposes new human-annotated datasets for complementary label learning (CLL). The motivation for the creation of such datasets is to relax the assumption of class-dependent instance-independent nature in current CLL datasets to the more general setting that not only depends on each instance, but also includes real-world label noise. The paper presents a detailed analysis on the characteristics of each dataset, including class imbalance and noise rate through transition matrices. Standard benchmarking on the newly-created datasets show a significant drop in terms of performance. In general, the newly-created datasets may provide a new benchmarking for CLL, linking towards a setting closer to the one in pratice.

**Audience:**

Yes

**Broader Impact Concerns:**

I do not have any comment on the broader impact statement of the paper.

**Claims And Evidence:**

Yes

**Requested Changes:**

Please make the approriate changes specified in the weaknesses.

**Strengths And Weaknesses:**

### Strengths
The paper proposes new datasets that are more challenging than existing ones due to their practical label noise and complementary class imbalance:
 - The paper provides the general description of each dataset.
 - The paper also presents the benchmarking of current CLL methods on the two datasets, demonstrating their challenging nature to encourage the community to investigate CLL in a more practical setting.

### Weaknesses
There are a few minor relating to the writing, such as:
- The two subfigures in Figure 1 are uneven. Please re-make them be in the same size. In addition, it is not helpful to set the y limit starting from 0. It is more helpful if ymin is set to 10,000 for Fig. 1(a) and 1,000 for Fig. 1(b).
- The display of the double quotes in the paragraph "Observation 2" on page 5 is not standard. In LaTEX, it can be done by: ``<something>''. In addition, the comma should be placed outside the double quotes. For example: "automobile," should be "automobile",

Beside that I do not see any major issues (I am not familiar to CLL in general).

---

> ### Author Response · Authors · 2025-04-14
>
> **Question 1**: The two subfigures in Figure 1 are uneven. Please re-make them be in the same size. In addition, it is not helpful to set the y limit starting from 0. It is more helpful if ymin is set to 10,000 for Fig. 1(a) and 1,000 for Fig. 1(b)
>
> **Answer 1**: Thank you for your valuable comment. We adjusted exactly *the same* y_min and the size of Figure 1 as your suggestion and updated it in the revised version.
>
> **Question 2**: The display of the double quotes in the paragraph "Observation 2" on page 5 is not standard. In LaTEX, it can be done by: ``<something>''. In addition, the comma should be placed outside the double quotes. For example: "automobile," should be "automobile",
>
> **Answer 2**: Thank you so much for your comment. We have re-checked and solved all relevant Latex syntax for that. We resubmitted the revised version.

---

### Decision · Action_Editor_Jucp · 2025-05-19

**Recommendation:** Accept as is

**Comment:**

The reviewers' scores were accept, accept, and leaning accept. All reviewers are satisfied with the rebuttal and the updated paper, and appreciate the realistic benchmark and analysis for complementary labels.

Although I recommend "Accept as is", it would be helpful if the paper incorporated the following suggestions:

- Cite Gebru et al. 2021 and note that their template/questions were used for Appendix H.
- In the Broader Impact Statement, the sentences "While our CLImage datasets do not belong to those. We suggest practitioners exercise caution when studying such applications." should be merged into a single sentence.
- The precise instructions that are provided to the annotators are missing from the paper. From the demo (mp4 file) in the anonymous github repository, I see the instruction `Choose any one "incorrect" label for this image`. It also appears that five example images are shown next to each of the four class options. It's possible I missed it, but these two pieces of information should be included in the paper in addition to the github readme.
- The answer to the question "Will the dataset be distributed under a copyright or other intellectual property (IP) license, and/or under applicable terms of use (ToU)?" in the Appendix only provided information about the original dataset. My understanding is that this question is asking the license about your new dataset (or labels). Please state the license for your neew dataset as well.
- Reference: "Ya Le and Xuan Yang. Tiny imagenet visual recognition challenge. Report of CS231N: Deep Learning for Computer Vision Course, 2015. Standford University." Correct imagenet to ImageNet and Standford to Stanford.
- Reference: "Learning multiple layers of features from tiny images" is a single-authored paper according to https://www.cs.toronto.edu/~kriz/learning-features-2009-TR.pdf
- Abstract: Regarding "Secondly, their evaluation has been limited to synthetic datasets.", I'm wondering if it would be more precise to say "synthetically labeled datasets" because a "synthetic dataset" sounds like the inputs/features are also synthetic, e.g., Gaussian distributions, two moons.

**Audience:**

Yes, researchers in the field of weakly supervised learning, benchmarking, and evaluation may be interested in the findings of this paper.

**Claims And Evidence:**

This paper proposes a protocol to collect complementary labels from human annotators, and as a result of the data collection, the paper provides the first real-world complementary-labelled datasets: CLCIFAR10, CLCIFAR20, CLMicroImageNet10, and CLMicroImagenet20. The claims are supported by accurate and clear evidence.

---

> ### Author Response · Authors · 2025-05-29
>
> Dear Action Editor,
>
> Thank you very much for your valuable and constructive feedback. We have made the following changes in the camera-ready version of our paper:
>
> 1. Cited Gebru et al. (2021) and noted that we used their template for our "Datasheets for Datasets."
> 2. Merged two related sentences for improved clarity.
> 3. Reviewed and corrected the instructions for "Annotation Task Design and Deployment on Amazon MTurk" in the README file.
> 4. Specified the license for our new datasets in the "Datasheets for Datasets."
> 5. Fixed a typo in one of the references.
> 6. Replaced "synthetic dataset" with "synthetically labeled datasets" throughout the abstract and the main text for accuracy.
> 7. Publicly released the code repository and datasets.
>
> Once again, we sincerely appreciate the comments from the reviewers and the Action Editor, which have greatly enhanced the quality of our work.
>
> Best wishes,